# In Silico Discovery of Potential Inhibitors Targeting the RNA Binding Loop of ADAR2 and 5-HT2CR from Traditional Chinese Natural Compounds

**DOI:** 10.3390/ijms241612612

**Published:** 2023-08-09

**Authors:** Emmanuel Broni, Carolyn Ashley, Miriam Velazquez, Sufia Khan, Andrew Striegel, Patrick O. Sakyi, Saqib Peracha, Kristeen Bebla, Monsheel Sodhi, Samuel K. Kwofie, Adesanya Ademokunwa, Whelton A. Miller

**Affiliations:** 1Department of Medicine, Loyola University Medical Center, Loyola University Chicago, Maywood, IL 60153, USA; 2Department of Molecular Pharmacology & Neuroscience, Loyola University Medical Center, Loyola University Chicago, Maywood, IL 60153, USA; 3Department of Biology, Loyola University Chicago, Chicago, IL 60660, USA; 4Department of Chemical and Biochemistry, College of Science, University of Notre Dame, Notre Dame, IN 46556, USA; 5Department of Chemistry, School of Physical and Mathematical Sciences, College of Basic and Applied Sciences, University of Ghana, Legon, Accra P.O. Box LG 56, Ghana; 6Department of Chemical Sciences, School of Sciences, University of Energy and Natural Resources, Sunyani P.O. Box 214, Ghana; 7Department of Biomedical Engineering, School of Engineering Sciences, College of Basic & Applied Sciences, University of Ghana, Legon, Accra P.O. Box LG 77, Ghana; 8Department of Biochemistry, Cell and Molecular Biology, West African Centre for Cell Biology of Infectious Pathogens, College of Basic and Applied Sciences, University of Ghana, Accra P.O. Box LG 54, Ghana; 9Department of Cognitive and Behavioral Neuroscience, Loyola University Chicago, Chicago, IL 60660, USA

**Keywords:** adenosine deaminases acting on RNA (ADAR), anti-ADAR2, natural products, cancer, depression, anxiety disorders, autism spectrum disorder (ASD), molecular docking, molecular dynamics simulation

## Abstract

Adenosine deaminase acting on RNA 2 (ADAR2) is an important enzyme involved in RNA editing processes, particularly in the conversion of adenosine to inosine in RNA molecules. Dysregulation of ADAR2 activity has been implicated in various diseases, including neurological disorders (including schizophrenia), inflammatory disorders, viral infections, and cancers. Therefore, targeting ADAR2 with small molecules presents a promising therapeutic strategy for modulating RNA editing and potentially treating associated pathologies. However, there are limited compounds that effectively inhibit ADAR2 reactions. This study therefore employed computational approaches to virtually screen natural compounds from the traditional Chinese medicine (TCM) library. The shortlisted compounds demonstrated a stronger binding affinity to the ADAR2 (<−9.5 kcal/mol) than the known inhibitor, 8-azanebularine (−6.8 kcal/mol). The topmost compounds were also observed to possess high binding affinity towards 5-HT_2C_R with binding energies ranging from −7.8 to −12.9 kcal/mol. Further subjecting the top ADAR2–ligand complexes to molecular dynamics simulations and molecular mechanics Poisson–Boltzmann surface area (MM/PBSA) calculations revealed that five potential hit compounds comprising ZINC000014637370, ZINC000085593577, ZINC000042890265, ZINC000039183320, and ZINC000101100339 had favorable binding free energies of −174.911, −137.369, −117.236, −67.023, and −64.913 kJ/mol, respectively, with the human ADAR2 protein. Residues Lys350, Cys377, Glu396, Cys451, Arg455, Ser486, Gln488, and Arg510 were also predicted to be crucial in ligand recognition and binding. This finding will provide valuable insights into the molecular interactions between ADAR2 and small molecules, aiding in the design of future ADAR2 inhibitors with potential therapeutic applications. The potential lead compounds were also profiled to have insignificant toxicities. A structural similarity search via DrugBank revealed that ZINC000039183320 and ZINC000014637370 were similar to naringin and naringenin, which are known adenosine deaminase (ADA) inhibitors. These potential novel ADAR2 inhibitors identified herein may be beneficial in treating several neurological disorders, cancers, viral infections, and inflammatory disorders caused by ADAR2 after experimental validation.

## 1. Introduction

ADAR2 is a part of a family of proteins called ADARs for adenosine deaminase acting on double-stranded RNA [1]. ADAR family proteins catalyze the hydrolytic deamination of adenosine to inosine (A-to-I), resulting in a high diversity of outcomes, although ADAR3 has no known catalytic activity [2,3]. Some of the impacts of A-to-I editing come from the direct changes in RNA sequences in coding regions. This RNA recording emerges because inosine is structurally similar to guanosine and can be interpreted by the cellular translational machinery as such. A popular and well-understood example of such editing includes glutamate receptor GRIA2 transcripts. In ADAR2 null mice, the mice suffer from epileptic-like seizures and die briefly after conception [4]. This phenotype is a repercussion of increased calcium permeability through the α-amino-3-hydroxy-5-methyl-4-isoxazole propionic acid (AMPA) receptor associated with a lack of RNA editing of the Q/R site in the GluA2 subunit by ADAR2 [5]. However, the majority of ADAR editing occurs in non-coding regions, including inverted Alu repeats [6,7]. The functionality and biological relevance of ADARs are far from being fully established. ADARs have been clearly implicated in a wide variety of complications such as viral infections [8], metabolic disorders [9], autoimmune and inflammatory diseases [10,11,12], several cancers, and neurological disorders [13,14]. These issues can be related to the dysregulation of ADAR RNA-editing activities that may generate codon alterations or splice site modulations, antagonize RNAi pathways, or interrupt miRNA processing [15,16]. The idea of ADAR proteins having RNA-editing independent activities through protein–protein interactions, sequestration, and other mechanisms also increases the potential roles of ADARs [17,18,19].

Some specific instances of ADAR2’s crucial roles include neurological disorders such as schizophrenia and major depressive disorder, roles in both tumor suppression and tumor aggravation, and its role in the innate immune system and viral infections. ADAR2 is known to edit multiple sites in the 5-hydroxytryptamine 2C receptor (5-HT_2C_R). Results from mice studies support that the editing of 5-HT_2C_R pre-mRNAs led to changes in the regulation of lipolysis and metabolism [20]. Variations in RNA editing efficiency at these sites led to symptoms that encompassed excessive anxiety and anti-depressive behaviors [20]. Dysregulated editing of 5-HT_2C_R pre-mRNA has been reported with psychiatric disorders, suggesting a role of ADAR2 in schizophrenia, autism, depression, and bipolar disorder [21,22,23,24]. ADAR2 plays conflicting roles in cancer. In some cancers ADAR2 activity may lead to proto-oncogenic effects, and in others ADAR2 acts as a tumor suppressor. ADAR2 plays an impactful role in GBM as it regulates a multitude of miRNAs in glioblastoma cells [25]. The reduction of ADAR2 editing on some miRNAs is involved in increasing tumor growth and migration in glioblastoma multiforme (GBM), which lead to a downregulation of their inhibition and thereby promoted their oncogenic activities [25,26]. In contrast, ADAR2 editing of certain transcripts in hepatocellular carcinoma (HCC) results in the regulation and suppression of oncogenic miRNAs. Thereby, in HCC, the upregulation of ADAR2 editing results in inhibited tumorigenicity [27,28]. ADAR2 plays specific roles in monitoring the self-compared to non-self RNA, which is key for regulating the innate immune response. ADAR2’s role in the innate immune response makes it a target of some viruses that use it to escape immune detection, but ADAR2 also has anti-viral abilities [29]. Borna disease virus (BoDV) is an RNA virus that utilizes ADAR2 editing of its own genomic RNA to avoid immune detection. Upon knockdown of ADAR2, editing of the virus was also reduced and resulted in an intense innate immune response [30].

With a wide variety of significant roles, ADARs make valuable drug targets for future therapies. Currently, there are no full models of ADAR proteins, but structures of valuable domains and motifs have been identified. Common domains between ADAR family proteins include multiple double-stranded RNA binding domains (dsRBD) and a singular catalytic deaminase domain (CDD) [31]. These domains are both involved in substrate selectivity. In ADAR editing there are two types of editing and certain preferences for editing sites that are impacted by RNA substrate structure and length. One type of editing is hypermutation, which is nonselective and rapidly deaminates adenosines that are commonly occurring in duplex RNA that is long and complimentary, and the other is a highly selective and accurate editing of duplex RNA that is short or broken up by bulges, mismatches, and loops [32,33]. ADARs also have nearest neighbor preferences that impact which adenosines will be edited. ADAR2 has a 5′ nearest neighbor preference for uracil followed by A > C > G and a 3′ nearest neighbor preference for G followed by C > U∼A [34]. A structural study of ADAR2 depicting the CDD of ADAR2 in complex with dsRNA suggests that the 3′ preference is impacted by the CDD [35]. However, another study of ADAR2’s dsRBD complexed to duplex RNA supports that the second dsRBD of ADAR2 impacts the 3′ nearest neighbor preference of ADAR2 [36]. The dsRBDs increase binding to duplex RNA in a sequence-independent manner [37]. Site specificity is predominantly impacted by the CDD. In a domain-swapping study, the CDD of ADAR1 and ADAR2 were exchanged and the proceeding substrate editing corresponded to the CDD [38]. The ability for the ADAR2 CDD to effect substrate specificity has been further explained by structures depicting the base-flipping mechanism of ADAR showing an RNA binding loop near the ADAR2 active site, whose steric clashes complimented the evidence for nearest neighbor preferences [35]. This RNA binding loop covers the amino acids of 454 to 479 with interactions made primarily to the sugar phosphate backbone of the RNA [35]. Conservation of this loop across ADAR2 sequences of other species support its significance, although this region deviates between ADAR family members potentially influencing substrate specificity [39]. Interestingly, experiments using high-throughput mutagenesis determined that of the 18 conserved residues within the ADAR2 5′ RNA binding loop only six required the original wild type amino acid to maintain RNA editing efficiency [39]. The six residues, Phe457, Asp469, His471, Pro471, Arg474, and Arg477, are involved in stabilizing the ordered conformation of the 5′ binding loop upon substrate binding and may provide additional dsRNA binding contacts [39]. The other 12 conserved residues in the RNA binding loop are not required for efficient editing, indicating that they play other key functional roles. It has been suggested that their conservation may correlate to their importance in protein–protein interactions for ADAR2 regulation or editing independent activities [39]. Overall, the six key residues of the 5′ binding loop of ADAR2 have been confirmed as necessary for efficient ADAR2 editing activity, and while ADAR1 and ADAR2 share significant sequence similarity in the majority of their CDDs, the 5′ binding loop represents an area of distinction between the two. Thus, indicating that targeting the 5′ binding loop of ADAR2 should affect the editing of ADAR2 substrates without affecting subsequential ADAR1 substrates.

This is incredibly important, as ADAR1 and ADAR2 can edit the same substrate at different editing sites such as in the 5-HT_2C_R. Within the five editing sites of the 5-HT_2C_R, ADAR1 edits sites A and B, ADAR2 edits site D, and sites C and E could be edited by either enzyme [40]. In a knockout study of ADAR2, it was shown that certain editing sites that relied on ADAR2, including the D site of 5-HT_2C_R, as well as the GluA2 Q/R site and the CYFIP2 K/E site, had reduced editing [40]. These ADAR2 knock out mice, in turn, did not develop enhanced ethanol intake or preference, even after chronic exposure to ethanol vapor, which did appear in the wild type mice. Therefore, ADAR2-dependent sites of the 5-HT_2C_R may contribute to alcohol intake. Additionally, as mentioned previously, altered RNA editing of the 5-HT_2C_R has been reported in several other complications, including major depression, depressed suicide victims, bipolar disorder, and schizophrenic patients, as well as in emotional influences such as anxiety and stress [21,41,42,43,44,45,46]. Thereby, the modulation of RNA editing of the 5-HT_2C_R may be therapeutic for these variable complications. However, the dearth of research on the screening of small molecule inhibitors that specifically target ADAR2 is surprising.

Pharmacoinformatic-based approaches are beneficial for analyzing and interpreting data related to drugs and drug action. These approaches aid in studying the properties of drugs and their interactions with biological systems, as well as in analyzing and interpreting large datasets generated by various sources, such as clinical trials, electronic health records, and drug databases. These, in turn, enable the identification of new, improved, safer, and more effective drug candidates, in addition to speeding up the process of repurposing therapeutic agents for existing and emerging diseases. Using molecular docking, this study virtually screened TCM compounds that target the RNA binding loop of the ADAR2 protein. The topmost compounds were subjected to molecular dynamics simulations and molecular mechanics Poisson–Boltzmann surface area (MM/PBSA) calculations to corroborate their affinity to the ADAR2 protein. The topmost compounds were further docked against the 5-HT_2C_R in order to investigate their multi-target inhibitory potential against both ADAR2 and 5-HT_2C_R. The biological activity of the shortlisted compounds were predicted using a Bayesian-based algorithm, prediction of activity spectra of substances (PASS) [47,48,49]. Furthermore, the pharmacokinetic and pharmacodynamics profiles of the shortlisted compounds were evaluated to determine their absorption, distribution, metabolism, excretion, and toxicity (ADMET) properties.

## 2. Results and Discussion

### 2.1. Selecting Binding Site

Previously, three potential binding sites of the human ADAR2 (hADAR2) protein were predicted [50] using Computed Atlas of Surface Topography of proteins (CASTp):IHP binding site [35,51] comprising of residues Ala389, Leu390, Asn391, Asp392, Ile397, Arg400, Arg401, Leu404, Tyr408, Gln500, Leu512, Thr513, Met514, Lys519, Arg522, Trp523, Val526, Gly527, Ile528, Gln529, Gly530, Ser531, Leu532, Leu533, Lys629, Leu632, Tyr658, His659, Lys662, Leu663, Tyr668, Gln669, Lys672, Phe676, Trp687, Val688, Glu689, Lys690, Pro691, Thr692, Gln694, and Asp695.RNA binding loop [35,39] comprising of residues Lys350, Val351, Gly374, Thr375, Lys376, Cys377, Ile378, Asn379, His394, Ala395, Glu396, Ile446, Thr448, Ser449, Pro450, Cys451, Gly452, Arg455, Ile456, Pro459, Lys483, Ile484, Glu485, Ser486, Gly487, Gln488, Gly489, Thr490, Leu511, Thr513, Cys516, Arg590, Lys594, and Ala595.A third plausible binding site lined by residues Ser458, His460, Glu461, Pro462, Ile463, Glu466, Pro467, Ala468, Asp469, Arg470, His471, His552, Asp554, and His555.

Recently, the IHP binding site [35,51] was virtually screened for potential anti-ADAR2 compounds, and nine compounds were shortlisted [50]. ADAR2 was also experimentally targeted using 8-azanebularine, nebularine, and coformycin. However, only 8-azanebularine was reported to inhibit ADAR2 reaction with an IC_50_ of 15 ± 3 mM [52]. Not much is reported about screening small molecule inhibitors against the ADAR2 RNA binding loop in literature. To the best of the authors’ knowledge, there is a dearth of studies which screen for small molecule inhibitors targeting the ADAR2 RNA binding site. Therefore, the RNA binding loop was selected for virtual screening in this study.

### 2.2. Molecular Docking of ADAR2

Molecular docking is the most popular and widely used computer-aided drug design technique for predicting the binding affinity and interactive modes of bioactive compounds and for performing receptor-based virtual screening studies [53,54]. Herein, AutoDock Vina embedded in PyRx was employed to virtually screen 8-azanebularine (known ADAR2 inhibitor) and natural compounds from the TCM library. The known inhibitor, 8-azanebularine, had a binding energy of −6.8 kcal/mol (Table 1). A previous study which shortlisted the top 10% of compounds after virtually screening against the ADAR2 IHP binding site reported the top compounds as having binding energies below −8.8 kcal/mol [50]. Herein, TCM compounds with binding energies below −9.5 kcal/mol were shortlisted.

A total of 310 compounds met this threshold (below −9.5 kcal/mol) and were selected for further analysis. The highest binding affinity to ADAR2 was observed for ZINC000095913861 with a binding energy of −12.0 kcal/mol, followed by ZINC000085996580, ZINC000070454467, and ZINC000042890265, with binding energies of −11, −10.9, and −10.6 kcal/mol, respectively. Compounds ZINC000039183320 and ZINC000085593577 had a binding energy of −10.5 kcal/mol, while ZINC000070454124, ZINC000103585067, and ZINC000014637370 had a binding energy of −10.2 kcal/mol. Also, compounds ZINC000013384051, ZINC000059586224, ZINC000070454074, ZINC000085530502, ZINC000085532258, ZINC000085532442, ZINC000095911347, and ZINC000095914813 all had a binding energy of −10.1 kcal/mol.

For the top 310 compounds, it was observed that their molecular weights ranged between 350 to 600 g/mol (Figure 1). All compounds which had binding energies lower than −11.0 kcal/mol had molecular weights greater than 500 g/mol (Figure 1). Only one compound (ZINC000014637370) with molecular weight less than 450 g/mol (408.49 g/mol) had a binding energy lower than −10.0 kcal/mol (−10.2 kcal/mol) (Figure 1). The binding energies of shortlisted compounds with large molecular weights suggest that the size and spatial characteristics of the RNA binding site may play a crucial role in facilitating ligand interaction and binding.

### 2.3. ADMET Prediction

The prediction of a molecule’s absorption, distribution, metabolism, excretion, and toxicity (ADMET) properties is of utmost importance in the drug discovery and development process. Although more accurate, traditional experimental approaches to assess these properties can be time-consuming, costly, and sometimes ethically challenging as compared to the computational approach. Notwithstanding, advancements in computational drug studies have provided powerful platforms that can predict these properties with remarkable accuracy. The 310 shortlisted compounds were subjected to ADMET profiling in order to select the ligands with the most desirable safety profiles (Table 2). Lipinski’s and Veber’s rules were applied to the compounds. A total of 136 compounds failed Lipinski’s rule, while 177 failed Veber’s rule.

The topmost compound, ZINC000095913861, with a binding energy of −12.0 kcal, passed both Lipinski’s and Veber’s rules with a TPSA of 94.56 Å^2^ (Table 2). However, ZINC000070450936 and ZINC000070454365, with binding energies of −11.2 and −11.1 kcal/mol, respectively, failed both Lipinski’s and Veber’s rules. ZINC000070450936 and ZINC000070454365 had TPSA values of 178.53 and 180.3 Å^2^, respectively. Overall, a total of 121 compounds passed both Lipinski’s and Veber’s rules and were shortlisted for further analyses. Of the 121 compounds, only five comprising ZINC000014637370, ZINC000085532375, ZINC000085547677, ZINC000085547700, and ZINC000101100339 were predicted as blood–brain barrier (BBB) permeants.

The 121 shortlisted compounds were further subjected to toxicity tests using OSIRIS DataWarrior [55], of which a total of 90 were shortlisted. Compounds that were predicted to have two or more toxicity risks were eliminated. Additionally, compounds with tumorigenic and mutagenic effects were eliminated since ADAR2 inhibition is implicated in certain cancers [56], including lung cancer [57]. Furthermore, ADAR2 inhibition may promote tumor growth since ADAR2 has been shown to suppress tumors [25,58,59]. Compounds with tumorigenic and mutagenic effects may be involved in promoting the growth of tumors and the spread of cancers during ADAR2 inhibition.

A total of nine compounds, including ZINC000013310993, ZINC000014686335, ZINC000085547677, ZINC000103559699, ZINC000085548190, ZINC000085567825, ZINC000085761575, ZINC000085976998, and ZINC000095914856, were predicted to be highly mutagenic, while 12 were predicted to have low mutagenicity. A total of 100 compounds were predicted to have no mutagenic effects. For tumorigenicity, three compounds, including ZINC000103578914, ZINC000103559699, and ZINC000085976998, were predicted as high; eleven (including ZINC000085594038, ZINC000085594040, ZINC000085594044, ZINC000085594057, ZINC000103527863, ZINC000103543220, ZINC000085547700, ZINC000085594065, ZINC000095914212, ZINC000085547677, and ZINC000085548190) were predicted as low; 109 were predicted to have none (Appendix A). A total of 76 compounds were predicted to have high reproductive effects while two (ZINC000085532375 and ZINC000013310993) were predicted as low, leaving 97 compounds with no reproductive effects. A total of 29 compounds were predicted to have high irritancy, while three (ZINC000103543220, ZINC000103578914, and ZINC000085548190) were low. A total of 89 compounds were predicted as non-irritants. In all, a total of 58 compounds were predicted as non-tumorigenic, non-mutagenic, and non-irritant, and had no reproductive effect (Appendix A).

The known inhibitor, 8-azanebularine, was also predicted as a non-tumorigenic, non-mutagenic, and non-irritant, and had no reproductive effect (Appendix A). The topmost compound, ZINC000095913861, with a binding energy of −12.0 kcal/mol, was predicted as non-tumorigenic, non-mutagenic, and non-irritant, but had a high reproductive effect (Appendix A). Compounds ZINC000085996580 and ZINC000070454467 with binding energies of −11.0 and −10.9 kcal/mol, respectively, also passed the toxicity risk filter. However, ZINC000085594057, which had a binding energy of −11.1 kcal/mol, was predicted to have no mutagenicity, low tumorigenicity, high reproductive effects, and high irritancy risks, and was thus eliminated (Appendix A). Also, two of the five compounds which were predicted as BBB permeants (ZINC000085547677 and ZINC000085547700) failed the toxicity risk filter (Appendix A).

### 2.4. ADAR2–Ligand Interaction Profiling

Protein–ligand interactions are crucial in drug discovery, molecular recognition, and understanding the mechanisms of biological processes. Therefore, visualizing and analyzing these interactions is essential as they provide insights into the binding modes and intermolecular contacts existing between proteins and small molecule ligands (Table 1, Figure 2, Appendix A). The known inhibitor, 8-azanebularine, interacted with the ADAR2 via hydrogen bonds with residues Thr375 (bond lengths of 2.94 and 3.14 Å), Ile484 (bond lengths of 2.82 and 3.2 Å), and Gly487 (bond length of 3.21 Å), and formed hydrophobic contacts with Lys376, Cys377, Ile378, His394, Lys483, and Glu485 (Figure 2a and Table 1). The pyrimidine ring of 8-azanebularine was observed to be involved in hydrophobic contacts with Cys377, Lys483, and Glu485; the hydroxides attached to the furan were involved in all five hydrogen bonds observed; the triazole formed hydrophobic interactions with Lys376 and Ile378 (Figure 2a,b). ZINC000095913861, which had the least binding energy (−12.0 kcal/mol), was observed to form two hydrogen bonds with Asn379 (2.92 Å) and Gly489 (2.99 Å) and hydrophobic contacts with Thr375, Lys376, Cys377, Ile378, His394, Arg455, Ile456, Lys483, Ile484, Glu485, Thr490, and Leu511 (Table 1, Appendix A).

ZINC000085996580 interacted with the ADAR2 via ten hydrogen bonds with residues Ile378 (3.01 Å), Arg455 (2.99 Å), Lys483 (3.17 Å), Ile484 (2.55 and 2.88 Å), Gly487 (3.05 Å), Leu511 (2.85 Å), Leu512 (3.11 Å), Thr513 (2.97 Å), and Arg590 (2.8 Å) (Table 1, Appendix A). It also formed hydrophobic interactions with Val351, Thr375, Lys376, Cys377, His394, Thr448, Cys451, Glu485, and Leu512. ZINC000070454467, which had a binding energy of −10.9 kcal/mol, was also observed to form two hydrogen bonds with His394 (3.15 Å) and Gly487 (2.91 Å), and hydrophobic bonds with Thr375, Lys376, Cys377, Lys483, Ile484, Ser486, Gln488, Gly489, Arg590, and Ala595 (Table 1, Appendix A). ZINC000042890265 (disulfuretin), an aurone derivative made up of two sulfuretins, was arranged in a cis isomeric form, establishing five hydrogen bonds with Cys377 (3.08 Å), Cys451 (3.15 Å), Gly452 (3.19 Å), Ser449 (2.7 Å), and Arg590 (3.05 Å), and 15 hydrophobic bonds with residues Lys350, Val351, Thr375, Lys376, His394, Glu396, Thr448, Pro450, Arg455, Ile456, Lys483, Ile484, Gly487, Gly489, and Thr490 (Table 1, Appendix A). ZINC000039183320 was also observed to form five hydrogen bonds with Cys377 (3.01 Å), Ile378 (3.19 Å), Asn379 (3.11 Å), Glu396 (2.77 Å), and Ser449 (3.14 Å), and 15 hydrophobic contacts with Val351, Thr375, Lys376, His394, Thr448, Pro450, Cys451, Arg455, Lys483, Ile484, Glu485, Gln488, Gly489, Arg590, and Ala595 (Table 1, Appendix A). ZINC000014637370, which has a benzopyrano pyran fused to a benzopyran (or chromene), interacted with Cys377 (3.34 Å), Asn379 (2.97 Å), Ser486 (3.31 Å), and Gly489 (3.19 Å) via hydrogen bonds and residues Thr375, Lys376, His394, Cys451, Lys483, Ile484, Glu485, Gly487, and Thr490 via hydrophobic bonds (Figure 2c,d). Two of the three oxygen atoms in the benzopyran-benzopyrano pyran core were involved in hydrogen bonds with Cys377 and Gly489, while the other two hydrogen bonds (Asn379 and Ser486) were formed with the oxygen bonded to the middle benzene in the benzopyrano pyran core (Figure 2c).

Analyzing the molecular interactions of multiple protein–ligand complexes help to identify critical residues involved in ligand binding as well as common binding patterns, which will aid in the design and optimization of ligands for therapeutic purposes. From the interaction profiles, residues Thr375, Lys376, Cys377, His394, Cys451, Arg455, Lys483, Ile484, Glu485, Gly487, Gln488, and Gly489 were common residues involved in ligand binding in the RNA binding site of the ADAR2.

### 2.5. Prediction of Biological Activity of the Selected Hit Compounds

In silico prediction of biological activity helps to prioritize compounds for further experimental testing. These predictions can be made using a variety of approaches, including structure-based methods, machine learning algorithms, and quantitative structure-activity relationship (QSAR) models. In silico prediction can also be used to evaluate the potential toxicity or therapeutic efficacy of a compound, as well as to identify novel compounds with desired biological properties. Herein, prediction of activity spectra of substances (PASS) was employed to determine the biological activity of the shortlisted compounds [47,48,60].

Compounds ZINC000085996580, ZINC000042890265, and ZINC000101100339 were predicted to be inhibitors of various deaminases, including blastcidin-S, pterin, creatinine, ornithine cyclodeaminase, glucosamine-6-phosphate, deoxycytidine, ATP, and cytosine deaminases. Since ADAR2 also belongs to the deaminase family, these compounds may possess anti-ADAR2 activity and are worthy of further experimental testing. Compounds ZINC000042890265 (Pa: 0.312 and Pi: 0.127), ZINC000085996580 (Pa: 0.270 and Pi: 0.188), and ZINC000014637370 (Pa: 0.255 and Pi: 0.213) were also predicted to be useful in dementia treatment. ZINC000101100339 was predicted as an antineurotic (Pa: 0.512 and Pi: 0.105) while ZINC000085532375 was predicted to be beneficial in neurodegenerative diseases treatment (Pa: 0.377 and Pi: 0.083). Also, ZINC000095913861 (Pa: 0.458 and Pi: 0.080), ZINC000014637370 (Pa: 0.336 and Pi: 0.168), and ZINC000042890265 (Pa: 0.278 and Pi: 0.239) were predicted as neurotransmitter uptake inhibitors. Selective serotonin reuptake inhibitors (SSRIs) such as sertraline and fluoxetine, which are neurotransmitter uptake inhibitors, exhibit beneficial effects in the treatment of depression, anxiety disorders, and certain forms of obsessive-compulsive disorder [61,62,63]. Sertraline and fluoxetine have also been found to be helpful in managing depression in individuals with epilepsy due to their ability to lower the risks of triggering seizures [64].

ZINC000042890265 and ZINC000101100339 were predicted to possess antialcoholic properties with Pa values of 0.228 and 0.220, respectively, with corresponding Pi values of 0.097 and 0.102. Abnormal ADAR2 editing of 5-HT_2C_R is implicated in increased alcohol intake [40], making ADAR2 a therapeutic target for alcoholism. ZINC000042890265 was further suggested to be useful in treating prion disease (Pa: 0.296 and Pi: 0.121), which is associated with increased RNA editing of FKRP and Rragd in mice. Compounds ZINC000042890265, ZINC000085593577, and ZINC000014637370 were predicted as antidyskinetic with Pa values of 0.460, 0.287, and 0.408, respectively, with corresponding Pi values of 0.076, 0.228, and 0.099. These compounds may prove useful in treating dyskinesia, which manifests in most neurologic disorders [65,66,67,68,69,70]. Compounds ZINC000042890265 (Pa: 0.725 and Pi: 0.013), ZINC000095913861 (Pa: 0.499 and Pi: 0.057), ZINC000085996580 (Pa: 0.595 and Pi: 0.033), ZINC000070454467 (Pa: 0.269 and Pi: 0.193), ZINC000039183320 (Pa: 0.514 and Pi: 0.053), ZINC000014637370 (Pa: 0.727 and Pi: 0.013), and ZINC000101100339 (Pa: 0.248 and Pi: 0.215) were predicted to possess anti-inflammatory properties.

Compounds ZINC000070454467, ZINC000095913861, ZINC000101100339, ZINC000014637370, ZINC000085996580, ZINC000039183320, ZINC000042890265, ZINC000085593577, and ZINC000085532375 were predicted as antineoplastics with Pa values of 0.995, 0.928, 0.793, 0.670 0.594, 0.533, 0.532, 0.317, and 0.290; and corresponding Pi values of 0.003, 0.005, 0.013, 0.031, 0.046, 0.062, 0.062, 0.143, and 0.158. They were also predicted as chemopreventive, chemoprotective, antimutagenic, anticarcinogenic, antimetastatic, and beneficial for treating prostate cancer, prostate disorders, proliferative diseases, and preneoplastic conditions. Furthermore, they were predicted to possess antileukemic properties and may demonstrate activity against various types of cancers, including breast, cervical, lung, ovarian, renal, gastric, colon, colorectal, uterine cancers, as well as carcinoma, melanoma, and sarcoma. Although ADAR2 is implicated in various cancers, it has been reported that ADAR2 is downregulated in esophageal squamous cell carcinoma (ESCC) [58]. These compounds may help in suppressing cancer growth and progression owing to ADAR2 inhibition. Furthermore, ZINC000042890265, ZINC000085996580, ZINC000014637370, and ZINC000101100339 were predicted as Pin1 inhibitors with Pa values of 0.663, 0.374, 0.324, and 0.252, respectively, with corresponding Pi values of 0.010, 0.103, 0.142, and 0.222. Increased Pin1-ADAR2 interactions have been shown to contribute to ADAR2 stability [71,72], and these interactions increase as neurons mature [71]. Pin1 is essential for editing GluA2 transcripts in cell lines and plays a role in regulating ADAR2 levels and its catalytic activity [72]. Pin1 is also implicated in promoting multiple cancer-driving processes [73], supporting the potential anti-cancer prediction of these compounds. ZINC000095913861 (Pa: 0.561 and Pi: 0.021), ZINC000085996580 (Pa:0.395 and Pi: 0.054), ZINC000039183320 (Pa: 0.194 and Pi: 0.174), ZINC000014637370 (Pa: 0.354 and Pi: 0.065), and ZINC000085532375 (Pa: 0.201 and Pi: 0.166) were predicted as dermatologic and may be beneficial in treating ulcers, pigmentation, and other skin-related issues in melanoma, as ADAR2 has been reported to play a crucial role in the stemness of melanoma and melanoma relapse [74].

Compounds ZINC000095913861 (Pa: 0.315 and Pi: 0.045), ZINC000085996580 (Pa: 0.293 and Pi: 0.060), ZINC000042890265 (Pa: 0.308 and Pi: 0.050), ZINC000039183320 (Pa: 0.284 and Pi: 0.066), ZINC000085593577 (Pa: 0.209 and Pi: 0.157), ZINC000014637370 (Pa: 0.283 and Pi: 0.067), ZINC000101100339 (Pa: 0.205 and Pi: 0.162), and ZINC000085532375 (Pa: 0.243 and Pi: 0.108) were predicted as RNA synthesis inhibitors. ZINC000095913861 (Pa: 0.224 and Pi: 0.066), ZINC000085996580 (Pa: 0.254 and Pi: 0.047), ZINC000042890265 (Pa: 0.189 and Pi: 0.104), ZINC000039183320 (Pa: 0.174 and Pi: 0.130), ZINC000085593577 (Pa: 0.200 and Pi: 0.089), ZINC000014637370 (Pa: 0.241 and Pi: 0.054), ZINC000085532375 (Pa: 0.205 and Pi: 0.084) were further predicted as RNA directed DNA polymerase inhibitors. Also, ZINC000042890265 (Pa: 0.364 and Pi: 0.084), ZINC000101100339 (Pa: 0.274 and Pi: 0.181) were predicted as RNA-directed RNA polymerase (RdRp) inhibitors. The compounds were also predicted to possess antiviral properties. BoDV, a non-segmented RNA virus [75,76], exploits the host’s ADAR2 throughout its life cycle to edit its genomic RNA in order to evade immune response [30]. A study showed that the knockdown of ADAR2 limits A-to-I editing of BoDV genomic RNA, leading to a strong host immune response [30]. This is not surprising, as both ADAR1 and ADAR2 regulate autoimmune responses [30,77]. Furthermore, BoDV is characterized by neurological disorders, including ataxia (affecting coordination, balance, and speech) and abnormal depressive behavior [76,78,79].

A structural similarity search via DrugBank revealed that ZINC000095913861 is similar to tanshinone I (Tan-I) with a score of 0.717. Tan-I, found in *Salvia miltiorrhiza*, has been shown to be effective in treating anti-inflammatory diseases, including mastitis [80], inflammation due to osteoarthritis [81], and neuro-inflammation [82]. Tan-I also possesses chemoprotective, chemopreventive, and anti-cancer properties against MCF-7 and MDA-MB-231 human breast cancer cells [83,84], colorectal cancer [85,86], hepatic carcinoma [84], gastric cancer [87], cervical cancer [88], ovarian cancer [89], and glioblastoma [90], among others. The broad-spectrum anti-cancer activity of Tan-I warrants the experimental testing of ZINC000095913861 on ADAR2 and cancer cell lines. Furthermore, Tan-I has been shown to possess neuroprotective properties, reverses cognitive and motor impairments, and enhances learning and memory in mice [91,92,93].

ZINC000070454467 is structurally similar to beta-escin (0.729), escin (0.729), and glycyrrhizic acid (0.703). The anti-cancer and anti-inflammatory properties of escin and beta-escin are recorded in literature [94,95,96]. Escin demonstrated anti-inflammatory activity in a mouse model of global cerebral ischemia, improved learning and memory recovery, and reduced hippocampal damage [97]. Escin was also reported to upregulate the expression of granulocyte-macrophage colony-stimulating factor (GM-CSF), which is known for its neuroprotective properties [97]. Oral administration of escin in a Parkinson’s disease mouse model inhibited neuro-inflammatory cytokine expressions in the substantia nigra [98]. The substantia nigra plays a crucial role in dopamine regulation, motor movements, learning, mood, and decision making [99,100]. Dysregulation of dopamine signaling has been associated with schizophrenia [101]. Also, glycyrrhizic acid inhibited kynurenine aminotransferase 2 (KAT2), with IC_50_ and Ki values of 4.51 ± 0.20 and 10.42 ± 1.62 μM, respectively. KAT2 catalyzes the conversion of kynurenine to kynurenic acid (KYNA) in brain tissues, and the accumulation of KYNA has been linked to schizophrenia [102].

ZINC000039183320 and ZINC000014637370 were predicted to be similar to naringin (scores of 0.746 and 0.704, respectively) and hesperidin (0.744 and 0.833, respectively). Furthermore, ZINC000014637370 is structurally similar to sakuranetin (0.862), naringenin (0.848), taxifolin (0.772), dihydromyricetin (0.772), and silibinin (0.709). Naringin and naringenin are known adenosine deaminase (ADA) inhibitors [103,104], with naringin inhibiting the deamination of cordycepin with Ki values of 58.8 and 168.3 μmol/L in mouse and human erythrocytes, respectively [103]. Taxifolin also inhibited the human ADA with IC_50_ of 400 μM [105]. The experimental testing of ZINC000039183320 and ZINC000014637370 as potential ADAR2 inhibitors is warranted since ADAR2 also belongs to the deaminase class of proteins. Hesperidin was shown to prevent and reverse ketamine-induced schizophrenia-like behaviors, including hyperactivity, social withdrawal, and cognitive impairment in mice [106]. Silibinin (10 µM) and naringenin (10 µM) were also shown to possess neuroprotective properties in zebrafish by reversing behavioral changes induced by bisphenol A (17.52 µM) [107]. Dihydromyricetin (DHM) is believed to counteract the intoxication effects of ethanol in mice and may be useful in treating alcohol use disorder [108]. DHM possesses neuroprotective activity and counteracts changes, including increased anxiety levels, reduced exploratory behaviors, and increased serum corticosterone levels and activation in NF-κB pathway, caused by socially isolating mice [109].

### 2.6. Molecular Dynamics Simulations

#### 2.6.1. Analyzing RMSD, RMSF, and Rg

The unbound ADAR2 protein and the ADAR2–ligand complexes were subjected to 100 ns molecular dynamics (MD) simulations. Understanding the molecular motion and conformational changes of macromolecules upon small molecule binding is germane to drug discovery [110,111]. MD simulations provide a platform to computationally study these atomic motions and fluctuations by using Newtonian physics approximations, taking into consideration the forces at play between bonded and non-bonded atoms [110,112]. Notwithstanding this, MD simulations have limitations, requiring lengthy simulation periods in order to correctly explain some dynamical properties and the paucity of mathematical descriptions of some of the physical and chemical forces that govern protein dynamics [113]. However, MD simulations have been shown to be comparable with experimental results and are very useful in drug discovery [113,114,115]. The root mean square deviation (RMSD), radius of gyration (Rg), and root mean square fluctuation (RMSF) were analyzed after the MD simulations.

Herein, the ADAR2-8-azanebularine complex demonstrated the greatest stability with an average RMSD of 0.196 ± 0.025 nm, followed by ADAR2–ZINC000101100339 and ADAR2–ZINC000014637370 complexes with average RMSD values of 0.201 ± 0.028 and 0.210 ± 0.023 nm, respectively (Figure 3). The ADAR2–ZINC000085532375, ADAR2–ZINC000042890265, ADAR2–ZINC000039183320, and ADAR2–ZINC000085593577 complexes also had average RMSD values of 0.211 ± 0.030, 0.230 ± 0.031, 0.238 ± 0.031, and 0.247 ± 0.030 nm, respectively (Figure 3). The unbound ADAR2 had an average RMSD of 0.217 ± 0.031 nm (Figure 3) [50] throughout the 100 ns, implying that the ADAR2 achieved higher stability when in complex with 8-azanebularine, ZINC000101100339, ZINC000014637370, and ZINC000085532375 (Figure 3). Although ADAR2 has loop regions, which may account for the slight increase in RMSD values [116], the RMSDs obtained herein are within the acceptable range, as RMSDs up to 3 Å (0.3 nm) are typical for most proteins after MD simulations [117].

The average Rg values of all ADAR2–ligand complexes showed that upon ligand binding, ADAR2 demonstrated lower, Rg implying greater compactness and stable folding compared to the unbound state [118]. The unbound ADAR2 had an average Rg of 2.070 ± 0.008 nm throughout the 100 ns simulation period while the ADAR2–ZINC000042890265 and ADAR2–ZINC000085532375 complexes had Rg values of 2.049 ± 0.006 and 2.054 ± 0.007 nm, respectively (Figure 4). The ZINC000042890265 demonstrated higher stability than all ten ligands, including IHP, which were previously shown to be good ADAR2 binders in the IHP binding site [50]. The ADAR2-8-azanebularine complex demonstrated comparable Rg (2.058 ± 0.007 nm) to that of ADAR2–ZINC000042890265 and ADAR2–ZINC000085532375 complexes (Figure 4). For the 100 ns simulation period, ADAR2–ZINC000014637370, ADAR2–ZINC000039183320, ADAR2–ZINC000085593577, and ADAR2–ZINC000101100339 complexes also had average Rg values of 2.058 ± 0.008, 2.064 ± 0.011, 2.065 ± 0.06, and 2.065 ± 0.007 nm, respectively (Figure 4).

For the RMSF analyses, residue indexes 380–390, 460–478, 492–515, 585–595, and 650–655 were observed to have high fluctuations for all the ADAR2–ligand complexes (Appendix A). On the other hand, at residue positions 352–359, 369–375, 394–397, 445–455, 482–486, and 512–560, very low fluctuations were observed (Appendix A), implying that these regions could be involved in strong interactions with the various ligands and thus have higher stability [119].

#### 2.6.2. Analyzing Snapshots and Hydrogen Bonds

The hydrogen bond interactions between the ADAR2 and each ligand were evaluated using “gmx hbond” (Figure 5). Snapshots at 25 ns intervals were also generated to visualize the position of the ligands at these time steps. The snapshots showed the 8-azanebularine was not stable in the RNA binding site of the ADAR2. The 8-azanebularine remained in the RNA binding pocket until 75 ns, where it moved to the region surrounded by residues Ser458, His460, Glu461, Pro462, Ile463, Glu466, Pro467, Ala468, Asp469, Arg470, His471, His552, Asp554, and His555, and remained there until the end of the simulation. This region was previously predicted via CASTp as a potential binding site of the ADAR2 protein [50]. According to “gmx hbond”, 8-azanebularine formed only one hydrogen bond with ADAR2 at times 25, 50, 75, and 100 ns. However, from the protein–ligand interaction maps, 2 H-bonds were observed at 25 ns with Ser458 (bond lengths of 2.96 and 3.09 Å) and only one was observed at times 50 [Glu466 (3.35 Å)], 75 [Asp503 (3.01 Å)], and 100 ns [Asp503 (2.66 Å)].

All the potential lead compounds except ZINC000085532375 were observed to bind stably in the RNA binding site throughout the simulation. Compound ZINC000085532375 moved away from the ADAR2 protein at 25 ns, implying a positive binding energy (repulsion), which could influence the binding affinity. However, at 50 ns ZINC000085532375 binds to ADAR2 in a different region, surrounded by residues Arg435, Leu436, Lys437, Val440, Pro571, Pro572, Leu573, Tyr574, and Thr575. It remained in this binding region until the end of the 100 ns simulation period. At times 50 and 100 ns, no hydrogen bonds with ADAR2 were observed. However, at 75 ns, ZINC000085532375 formed a hydrogen bond with Leu573 of length 3.00 Å, which was also predicted via “gmx hbond”.

Compound ZINC000042890265 formed the highest number of hydrogen bonds with ADAR2 (7) at 17 ns. At 25 ns, hydrogen bonds with Cys377 (3.14 and 3.15 Å), Arg455 (2.9 Å), Gln488 (2.99 Å), and Gly489 (2.89 Å) were observed, although “gmx hbond” predicted 3. At 50 ns, only one hydrogen bond with Cys377 (3.17 Å) was maintained, while forming new bonds with Lys350 (2.69 Å), Ser449 (2.76 and 3.29 Å), and Gln591 (2.82 Å). At 75 ns, four hydrogen bonds were formed: two with Ser449 (2.76 and 2.90 Å) and one each with Arg455 (2.78 Å) and Gln488 (2.75 Å). At the end of the simulation, three hydrogen bonds were formed with residues Ser449 (2.68 Å), Arg455 (2.99 Å), and Glu485 (2.45 Å), although only one was predicted via “gmx hbond”. The number of hydrogen bonds formed by the ADAR2–ZINC000042890265 complex makes ZINC000042890265 an interesting candidate to probe further since ligand binding and activity have been shown to be influenced by the formation of multiple hydrogen bonds [120,121]. Another complex worth mentioning is ADAR2–ZINC000085593577. ZINC000085593577 was predicted via “gmx hbond” to form 1, 1, 2, and 3 hydrogen bonds with ADAR2 at times 25, 50, 75, and 100 ns, respectively (Figure 5). However, from the interaction maps, one [Gly489 (2.99 Å)], three [Asn391 (2.71 Å) and His394 (3.13 and 3.20 Å)], three [Asn391 (2.75 Å), His394 (2.82 Å), and Gly487 (2.77 Å)], and four [Asn391 (3.32 Å), His394 (2.84 Å), Gln488 (3.13 Å), and Gly489 (2.91 Å)] were formed at 25, 50, 75, and 100 ns, respectively.

### 2.7. MM/PBSA Calculations for the ADAR2–Ligand Complexes

#### 2.7.1. Analyzing Binding Free Energy

The molecular mechanics/Poisson–Boltzmann surface area (MM/PBSA) method is a computational technique that is used to calculate the binding free energy of a small molecule to a protein [122,123,124,125]. The binding free energy is a measure of the strength of the interaction between the small molecule and the protein, and it is an important factor in drug discovery, as molecules with a favorable binding free energy are more likely to be effective as drugs. The g_mmpbsa tool was used to perform the MM/PBSA calculations of the protein–ligand complexes using the complete 100 ns simulation trajectories [122]. The binding free energy, van der Waals (vdW), electrostatic, polar solvation, and solvent accessible surface area (SASA) energies were computed using this method (Table 3).

The vdW energies of all the complexes ranged from −49.600 (8-azanebularine) to −224.512 kJ/mol (ZINC000085593577) (Table 3), while the electrostatic energies ranged from 310.967 (8-Azanebularine re-run) to −81.409 kJ/mol (ZINC000042890265). ADAR2–ZINC000042890265 and ADAR2–ZINC000085532375 complexes demonstrated the highest and least polar solvation energies, with values of 174.426 and 18.341 kJ/mol, respectively (Table 3). The vdW, electrostatic, and polar solvation energies were major contributors to the binding free energies of the ADAR2–ligand complexes than the SASA energies. A similar trend was observed for ligands which bound in the IHP binding site of ADAR2 [50]. The SASA energies ranged from −24.866 (ADAR2–ZINC000042890265) to −7.374 kJ/mol (ADAR2-8-Azanebularine), with the ADAR2-8-azanebularine re-run demonstrating a SASA energy of −13.942 kJ/mol (Table 3). ADAR2–ZINC000085593577, ADAR2–ZINC000014637370, ADAR2–ZINC000101100339, ADAR2–ZINC000039183320, and ADAR2–ZINC000085532375 complexes had SASA energies of −24.585, −22.854, −16.419, −15.755, and −9.868 kJ/mol, respectively (Table 3). This implies that ZINC000042890265, ZINC000085593577, ZINC000014637370, ZINC000101100339, and ZINC000039183320, with lower SASA energy values than 8-azanebularine and ZINC000085532375, are situated within a hydrophobic environment at the active site of ADAR2, resulting in reduced exposure to water molecules from the surrounding physiological medium [126].

The known ADAR2 inhibitor, 8-azanebularine, had a binding free energy of 246.374 kJ/mol (Table 3). A re-run of the MM/PBSA calculation after another MD simulation of the ADAR2-8-azanebularine complex revealed a binding free energy of 310.779 kJ/mol. These two different simulation runs show that 8-azanebularine does not form strong bonds with ADAR2. This is not surprising, as 8-azanebularine was experimentally reported to inhibit ADAR2 reaction with a weak IC_50_ of 15 ± 3 mM [52]. Generally, the MM/PBSA method is most accurate for compounds with IC_50_ values in the micromolar and nanomolar ranges [127]. All the potential lead compounds, except ZINC000085532375, had favorable binding free energy with the ADAR2 protein. Compound ZINC000085532375 was observed to have a binding free energy of 228.669 kJ/mol (Table 3) and was thus eliminated. Both 8-azanebularine and ZINC000085532375 demonstrated very high electrostatic energies of 227.826 and 289.884 kJ/mol which influenced their binding free energy values. Both ligands were also observed to be unstable in the RNA binding pocket during the MD simulations. Compound ZINC000014637370, on the other hand, demonstrated the highest binding affinity to the ADAR2 protein with a binding energy of −174.911 kJ/mol, followed by ZINC000085593577, ZINC000042890265, ZINC000039183320, and ZINC000101100339 with binding free energies of −137.369, −117.236, −67.023, and −64.913 kJ/mol, respectively (Table 3).

#### 2.7.2. Analyzing Per-Residue Energy Contributions

Per-residue energy decompositions were performed on each complex after the MM/PBSA computations to investigate the energy contributed by each amino acid (Figure 6 and Appendix A) [123,125]. From the interaction maps, residues Thr375, Lys376, Cys377, His394, Cys451, Arg455, Lys483, Ile484, Glu485, Gly487, Gln488, and Gly489 were observed to be involved in most of the ADAR2–ligand interactions (Table 1). While molecular docking has demonstrated an ability to produce conformations resembling experimentally determined protein–ligand complex structures [128,129], it is more appropriate to analyze the protein–ligand interactions from MD simulations. Molecular docking can provide a conformation/pose that serves as a favorable starting point for conducting molecular dynamics simulation-based investigations [128]. Docking poses need to be verified by MD simulations and the interactions can be better understood from simulations than docking poses [130]. Moreover, the MM/PBSA method considers the energetics of protein–ligand interactions, including the dynamic behavior of the system during molecular dynamics simulations.

The per-residue energy decomposition analyses revealed a wide range of energy contributions among the ADAR2 residues, highlighting their varying roles in stabilizing the ADAR2–ligand complexes. Notably, key residues Lys350, Cys377, Cys451, Arg455, Ser486, Gln488, and Arg510 exhibited significant energy contributions, suggesting their crucial involvement in ligand recognition and binding. Glu396, on the other hand, was observed to contribute high energy which is unfavorable for ligand binding. Residues Cys377, Cys451, Arg455, and Gln488 were predicted via both docking and MM/PBSA as important for ligand binding in the RNA binding loop. These results further corroborate the predictions obtained via molecular docking. Lys350 (Lys867 in ADAR1), which is conserved across ADAR proteins, is positioned to face the major groove of the RNA and is in close proximity to the phosphate group adjacent to the flipped-out 8-azanebularine moiety, although Lys350 is not involved in direct interactions with the RNA substrate in the ADAR2–RNA structure [131]. Cys451 and Glu396 are involved in coordinating the zinc ion and ensuring its presence in the catalytic site [16,51]. In the hADAR2, Arg455 and Arg376 are reported to form symmetrical interactions with the phosphate groups located upstream and downstream of the flipped base, helping to anchor the flipped base in a proper orientation for catalytic activity [132]. Therefore, Arg455 mutation to alanine disrupts the symmetrical interactions and reduces steric hindrance on one side, potentially weakening substrate binding [132].

For the ADAR2–ZINC000042890265 complex, seven residues contributed energies above +5 or below −5 kJ/mol. Lys350 (−6.1117 ± 0.6947 kJ/mol), Lys376 (−11.2861 ± 0.5054 kJ/mol), Cys451 (−6.6857 ± 0.1919 kJ/mol), Arg455 (−8.3875 ± 0.6062 kJ/mol), and Gln488 (−7.4301 ± 0.4431 kJ/mol) contributed favorably to ZINC000042890265’s binding, while Glu396 (10.1660 ± 0.7664 kJ/mol) and Glu485 (11.4984 ± 0.5526 kJ/mol) contributed positive energy values above the +5 kJ/mol threshold (Appendix A). Other residues worth mentioning include Arg349 (−4.340 ± 0.1412 kJ/mol), Arg481 (−3.4924 ± 0.1009 kJ/mol), Ser486 (−3.1868 ± 0.1074 kJ/mol), Gly487 (4.1502 ± 0.2038 kJ/mol), Arg510 (−3.3549 ± 0.1858 kJ/mol), Glu588 (4.2705 ± 0.1095 kJ/mol), Pro592 (−4.6041 ± 0.3091 kJ/mol), and Lys594 (−3.8252 ± 0.7079 kJ/mol). For the ADAR2–ZINC000039183320 complex, only Phe457 contributed energy above the threshold with a value of −8.1866 ± 0.6182 kJ/mol (Appendix A). For ADAR2–ZINC000101100339, Cys377 (−5.7736 ± 0.2787 kJ/mol) and Arg510 (−6.4314 ± 0.1461 kJ/mol) contributed significantly to ZINC000101100339 binding in the RNA binding loop (Appendix A). For the ADAR2–ZINC000014637370 complex, Cys377 (−5.4945 ± 0.2346 kJ/mol), Arg455 (−8.5618 ± 0.3376 kJ/mol), Ser486 (−6.6641 ± 0.1840 kJ/mol), Gln488 (−13.2511 ± 0.4194 kJ/mol), and Gly489 (−5.1627 ± 0.2240 kJ/mol) contributed favorable energies, while Glu396 (6.0735 ± 0.7359 kJ/mol) was unfavorable for ligand binding (Appendix A). For ADAR2–ZINC000085593577, Lys350 (−5.6598 ± 0.2957 kJ/mol), Cys451 (−5.2814 ± 0.1461 kJ/mol), Arg455 (−7.2607 ± 0.3821 kJ/mol), and Leu511 (−5.8458 ± 0.2051 kJ/mol) contributed favorably, while Glu396 (9.5031 ± 1.2550 kJ/mol) contributed unfavorable energy to ZINC000085593577 binding (Appendix A).

For ADAR2–ZINC000085532375 (Appendix A) and ADAR2-8-azanebularine (Appendix A) complexes, several residues contributed significant energies due to the dynamic nature of the ligand binding process. During the course of the 100 ns MD simulation, 8-azanebularine and ZINC000085532375 exhibited mobility and explored different binding cavities of ADAR2. This movement led to interactions with multiple residues at different stages of the simulation, resulting in varying energy contributions. For the ADAR2-8-azanebularine complex, a total of 38 residues contributed favorable energies while 52 residues contributed energies above +5 kJ/mol. For the ADAR2–ZINC000085532375 complex, a total of 39 ADAR2 residues contributed energies below −5 kJ/mol, while 52 residues contributed above +5 kJ/mol. The instability of 8-azanebularine in the RNA binding site and binding site hopping could be responsible for its weak IC_50_ value previously reported [52].

### 2.8. Re-Docking of Top Compounds against the 5-HT_2C_ Receptor

Since the aberrant ADAR2 editing of the 5-HT_2C_ receptor causes major depressive disorder (MDD), suicidal behavior, anxiety disorders, and schizophrenia, this study sought to screen the top compounds against the 5-HT_2C_R to determine their potential binding affinities. All the compounds were observed to firmly dock in the ritanserin binding site. Ritanserin had a binding energy of −12.7 kcal/mol, the same as previously reported [50]. The least binding energy was observed for ZINC000095913861 (−12.9 kcal/mol) (Table 4), the same compound with the least binding energy to ADAR2 (−12.0 kcal/mol) (Table 1). ZINC000014637370, which demonstrated the strongest binding to ADAR2 (−174.911 kJ/mol) from the MM/PBSA calculations, was observed to have a binding energy of −10.9 kcal/mol. ZINC000085593577, which had a binding free energy of −137.369 kJ/mol with ADAR2, had a binding energy of −11.4 kcal/mol with 5-HT_2C_R. Experimental testing is required to determine the potential polypharmacologic activities against ADAR2 and 5HT_2C_R.

### 2.9. Provenance of Potential Lead Compounds

Various databases, including ZINC15 [133], PubChem [134,135], ChEMBL [136,137,138], and LOTUS [139], as well as existing literature, were searched to identify the sources of the five potential lead compounds. The existing literature was also investigated for the pharmacological activities of the plant sources. Structures of the shortlisted compounds and known inhibitors used in this study are provided (Figure 7).

ZINC000042890265 (disulfuretin), an aurone derivative, can be found in *Cotinus coggygria* (known as “smoke tree”) [140,141]. Disulfuretin demonstrated strong antioxidative activity in a 2,2-diphenyl-1-picrylhydrazyl (DPPH) assay with an IC_50_ of 9.7 μg/mL [140]. The ethyl acetate (EtOAc) and ethyl alcohol (EtOH) extracts of *C. coggygria* have also being reported to possess strong antioxidative properties [140,141]. Furthermore, *C. coggygria* extracts have demonstrated cytotoxic activities against glioblastoma cells, Hep-G2, MCF-7, A549, and HCT116 with IC_50_ values of 45.68 ± 2.26, 65.47 ± 2.48, 48.23 ± 2.30, 32.40 ± 2.02, and 33.13 ± 2.03 μg/mL, respectively [141]. This makes ZINC000042890265 an interesting candidate to test for its potential anti-cancer activity against brain cancer, hepatocellular carcinoma, breast cancer, human non-small cell lung cancer, and human colorectal carcinoma. The anti-inflammatory, anti-microbial, hepatoprotective, and antidiabetic activity of *C. coggygria* have been highlighted in the literature [142,143,144]. ZINC000039183320 (neocalyxin A), a diarylheptanoid derivative, is also found in the seeds of *Alpinia blepharocalyx* and *A. roxburghii* [145,146,147]. EtOH extracts of the seeds of *A. blepharocalyx* possess antiproliferative activity against both human HT-1080 fibrosarcoma and murine colon 26-L5 carcinoma cells, with neocalyxin A showing ED_50_ values 10.7 and >100 μM against the two cell lines, respectively [145]. However, neocalyxin A demonstrated 43% inhibition of the murine colon 26-L5 carcinoma cells at 50 μg/mL [145].

Compound ZINC000101100339 (qingdainone) can be found in *Isatis tinctoria*, also known as banlangen (BLG) [148]. BLG is used in traditional Chinese medicine to prevent and treat respiratory virus infections such as influenza [149,150]. Extracts of *I. tinctoria* have shown strong anti-inflammatory activity [151,152]. Also, the hydroalcoholic leaf extract of *I. tinctoria* was shown to possess anti-depressive properties and can reduce stress-induced behavioral disorders in mice [153], making qingdainone an interesting candidate to test for its neuroprotective properties. No source information was found for compounds ZINC000014637370 (PubChem CID: 163005156) and ZINC000085593577 (PubChem CID: 162813068).

## 3. Materials and Methods

### 3.1. Protein and Ligands Preparation

The hADAR2 structure (PDB ID: 5ed2) was obtained from the Research Collaboratory for Structural Bioinformatics (RCSB) Protein Data Bank [154,155]. This structure consists of a mutant E488Q of the hADAR2 deaminase domain complexed with inositol hexakisphosphate (IHP) and a double-stranded ribonucleic acid (dsRNA) [35]. To prepare the structure for molecular docking, the dsRNA, IHP ligand, and zinc atoms bound to the hADAR2 protein were removed using PyMOL (version 2.3.0). The resulting structure was processed using the Protein Preparation Wizard in Maestro (Schrödinger, LLC, New York, NY, USA) and optimized using the OPLS4 force field to address any steric hindrance and optimize protein energies [156]. For the screening of potential ligands, a total of 35,161 natural products from the Traditional Chinese Medicine (TCM) database were obtained from TCM@Taiwan, which is the largest non-commercial TCM database and a catalog of the ZINC15 database [133,157]. The compounds were pre-filtered based on molecular weight using OSIRIS DataWarrior 5.5.0 [55], similar to previous studies [130,158]. Compounds with molecular weights below 150 g/mol or above 600 g/mol were excluded, resulting in a final set of 25,196 compounds within the specified range. 8-azanebularine was extracted from PubChem with CID 10106291 and was used as a standard or control in the study.

### 3.2. Molecular Docking Studies

AutoDock Vina embedded in PyRx version 0.9.2 was used for the molecular docking process. The TCM compounds and the known inhibitor were virtually screened against the ADAR2 protein targeting the RNA binding site using grid box dimensions of 27.319 × 30.006 × 33.189 Å^3^ and the protein centered at x = 18.670 Å, y = 38.261 Å, and z = 77.541 Å. Prior to molecular docking, ligand structures in structure-data file (sdf) format were subjected to energy minimization using the UFF force field and 25,189 compounds were successfully converted to AutoDock’s Protein Data Bank, Partial Charge, and Atom Type (PDBQT) format. An exhaustiveness of 8 was set for the molecular docking process. After docking, the pose with the lowest binding energy (highest binding affinity) was selected for each compound since AutoDock Vina generates up to nine conformations for each ligand. The docking results were analyzed and ligands with binding energies below −9.5 kcal/mol were selected for further analysis.

### 3.3. ADMET Profiling

The ADMET profiles of top compounds with favorable binding energies were predicted using SwissADME [159]. The SMILES format of each compound was used as inputs for SwissADME. Lipinski and Veber’s rules were used as filters to shortlist compounds with relatively safe ADME profiles. While Lipinski’s rule allows up to one violation of the four criteria, Veber’s rule requires that compounds with good oral bioavailability should not violate any of the two criteria. For a compound to pass Veber’s rule, it should not have more than ten rotatable bonds and its topological polar surface area (TPSA) should not exceed 140 Å^2^ [160]. Lipinski’s rule of five expects safe drugs to have not more than 5 hydrogen bond donors; less than 10 hydrogen bond acceptors; molecular mass should not exceed 500 Da; and an octanol–water partition coefficient (logP) not more than 5 [161,162]. OSIRIS DataWarrior 5.5.0 was also employed to predict toxicity risks of the shortlisted compounds [55]. DataWarrior was used to evaluate the mutagenic, tumorigenic, reproductive effect, and irritant risks of compounds. After uploading the sdf format of the top compounds into DataWarrior, the “Chemistry” tab, “From Chemical Structure”, and then “Calculate Properties” were selected. Under the “LE, Tox, Shape” tab, “Mutagenic”, “Tumorigenic”, “Reproductive Effects”, and “Irritant” were selected in order to predict the toxicity risks.

### 3.4. Visualizing ADAR2–Ligand Interactions

The ADAR2–ligand interaction profiles involving the top compounds were determined using LigPlot+, a widely used tool for visualizing and analyzing protein–ligand interactions [163]. Each ADAR2–ligand complex was uploaded, and under the “LIGPLOT” tab the ligand was selected prior to clicking the “Run” button. LigPlot+ generates 2D schematic diagrams that depict the specific hydrogen bonds and hydrophobic interactions between the protein and ligand. The LigPlot+ analysis provides valuable insights into the binding mode and binding interactions between the protein and ligand, aiding in the interpretation of their functional significance.

### 3.5. Biological Activity Prediction of Shortlisted Compounds

The prediction of activity spectra of substances (PASS) was used to predict the likely pharmacological activity of shortlisted compounds (available at http://www.way2drug.com/passonline/predict.php, accessed on 23 May 2023) [47,48,60]. PASS utilizes structural information and statistical analysis to estimate the biological activity profiles of substances. By analyzing the chemical structure and physicochemical properties of the compounds, PASS generates predictions regarding their potential activities across a wide range of pharmacological targets and therapeutic areas. These predictions provide valuable insights into the possible biological activities of the compounds, enabling the identification and prioritization of promising candidates for further experimental validation. The SMILES format of the compounds was used as inputs for the activity prediction. The “Pa > Pi” filter was selected to only display results that had a probability of activity greater than the probability of inactivity.

### 3.6. Molecular Dynamics Simulations Study

GROningen MAchine for Chemical Simulations (GROMACS) version 5.1.5 was employed for molecular dynamics (MD) simulations of the unbound ADAR2 protein and selected ADAR2–ligand complexes [164,165]. The ligand topologies of the compounds were generated using LigParGen [166] for the OPLS force field. To solvate each ADAR2–ligand complex, a cubic box was employed, and the “TIP4P” water model and the OPLS/AA force field were utilized [167,168]. In order to neutralize the system charges, sodium or chlorine ions were added to the solvated complexes. The systems underwent initial equilibration in the constant number, constant-volume, and constant-temperature (NVT) ensemble, as well as in the isothermal-isobaric or constant number, constant-pressure, and constant-temperature (NPT) ensemble before the 100 ns MD simulation. After the MD simulations, the root mean square deviation (RMSD), radius of gyration (Rg), and root mean square fluctuation (RMSF) of each system were analyzed. Additionally, the hydrogen bond count was monitored throughout the simulation for each system. Snapshots of each complex were generated at 25 ns intervals (time step = 0, 25, 50, 75, and 100 ns).

### 3.7. Molecular Mechanics Poisson-Boltzmann Surface Area Calculation

After MD simulations, the resulting complexes were subjected to MM/PBSA calculations to determine the energy terms (vdW, electrostatic, polar solvation, SASA, and binding free energies) using the g_mmpbsa tool [122]. The MM/PBSA calculations were performed using the complete 100 ns simulation trajectories which have time steps of 1 ns. Also, the energy contribution of each ADAR2 residue was calculated for each of the ADAR2–ligand complexes. The decomposition of the free energy at the per-residue level helps to identify key residues, contributing significantly to the overall binding affinity.

### 3.8. Re-Docking Hit Compounds against the 5-HT_2C_R

The top compounds were virtually against the 5-HT_2C_R structure (PDB ID: 6BQH) [169] using AutoDock Vina embedded in PyRx. Ritanserin, which was bound to the 6BQH structure [169], was extracted and used as the control. Amino acid residues, which are in contact with ritanserin after visualizing the 5-HT_2C_R-ritanserin interaction map (Trp130, Asp134, Val135, Ser138, Thr139, Ile142, Phe320, Trp324, Phe327, Phe328, Leu350, Asn351, Val354, and Tyr358), were selected via PyRx, and the docking grid box was set to cover these residues. The grid box had dimensions of 23.8457393391 × 21.0061927432 × 22.651986164 Å^3^, and the protein was centered at x = 37.2017597536 Å, y = 29.7121195973 Å, and z = 53.165999618 Å.

## 4. Conclusions

Although aberrant ADAR2 editing is implicated in several disorders, including neurological disorders, cancers, viral infections, alcoholism, metabolic disorders, and inflammatory disorders, very few attempts have been made to identify small molecule inhibitors targeting the ADAR2. This study employed molecular docking and dynamics simulations to shortlist compounds from a total of 35,161 traditional Chinese medicine by targeting the RNA binding loop of the ADAR2 protein. Shortlisted compounds were further subjected to MD simulations and MM/PBSA calculations, and they demonstrated higher binding affinity to ADAR2 than the control, 8-azanebularine. By decomposing the overall binding free energy into individual residue contributions, residues Lys350, Cys377, Glu396, Cys451, Arg455, Ser486, Gln488, and Arg510 were identified as key residues that play critical roles in stabilizing the protein–ligand complexes. The observed interactions and energy contributions provide valuable insights into the molecular mechanisms governing ligand recognition and binding. The findings reported herein will pave the way for further investigations and rational design of novel ligands targeting the RNA binding loop of the hADAR2 protein. A total of five potential ADAR2 inhibitors comprising ZINC000042890265, ZINC000039183320, ZINC000101100339, ZINC000014637370, and ZINC000085593577 were identified in this study. Safety and toxicity predictions also suggested that the compounds possess insignificant toxicities. Re-docking the top compounds against the serotonin 2C receptor also showed that they possess remarkably strong binding affinity to 5-HT_2C_R in the same binding site as ritanserin, a known 5-HT_2C_R inhibitor. This additional finding indicates that the potential versatility of the identified potential compounds as multi-target agents, capable of treating diseases associated with aberrant RNA editing as well as serotonin receptor-related disorders. The significance of this study lies in the identification of novel compounds from traditional Chinese medicine library, offering new possibilities for treating a wide range of disorders, including neurological disorders, cancers, viral infections, alcoholism, metabolic disorders, and inflammatory disorders, all of which are linked to aberrant RNA editing. The insights gained from this research will pave the way for further experimental validations and investigations in vitro and in vivo to confirm the efficacy and safety of the potential ADAR2 inhibitors. The findings represent a significant step forward in the field of drug discovery targeting ADAR2, and the potential therapeutic benefits of these compounds warrant further exploration.

## Figures and Tables

**Figure 1 ijms-24-12612-f001:**
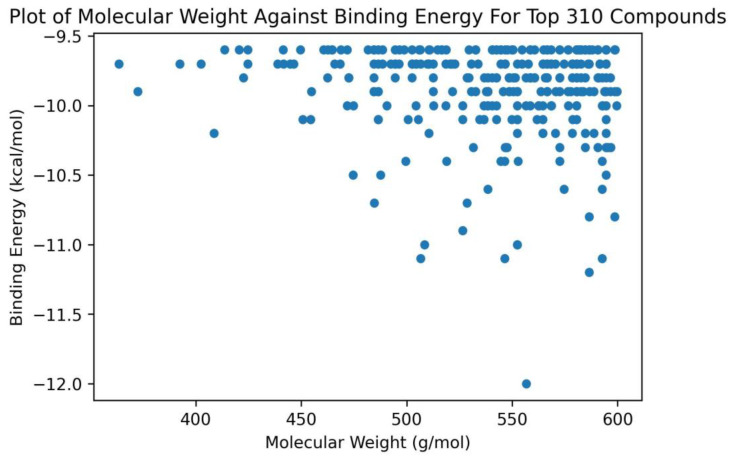
A plot of molecular weight against the binding energies of the top 310 compounds after docking with ADAR2. Only compounds with molecular weights above 450 g/mol had binding energies below −10.5 kcal/mol.

**Figure 2 ijms-24-12612-f002:**
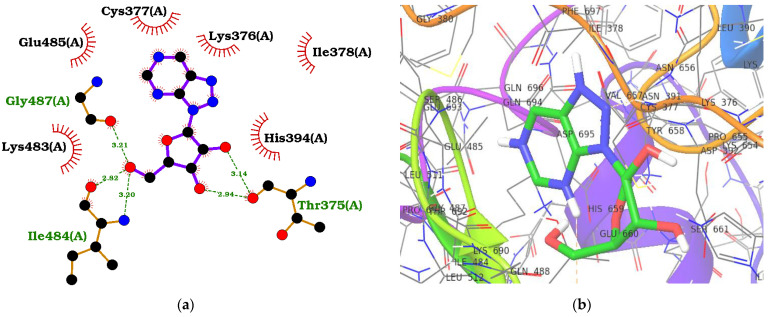
Protein–ligand interaction maps for ADAR2 complexed with 8-azanebularine (**a**,**b**) and ZINC000014637370 (**c**,**d**). For the 2D interaction maps (**a**,**c**), the ligand is colored violet, hydrogen bond lengths are labelled green, and hydrophobic contacts are shown as red arcs with spokes towards the ligand. For the 3D profiles (**b**,**d**), the ligands are shown as sticks while the protein is represented as ribbons with lines.

**Figure 3 ijms-24-12612-f003:**
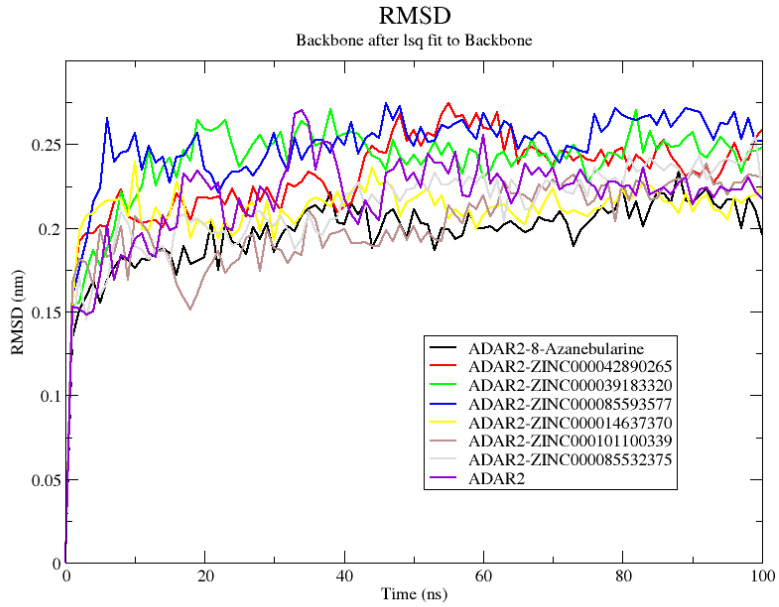
Root mean square deviation (RMSD) plot of the unbound ADAR2 protein and ADAR2–ligand complexes throughout the 100 ns MD simulations. ADAR2–8-azanebularine, ADAR2–ZINC000101100339, ADAR2–ZINC000014637370, and ADAR2–ZINC000085532375 complexes demonstrated lower average RMSDs than the unbound ADAR2.

**Figure 4 ijms-24-12612-f004:**
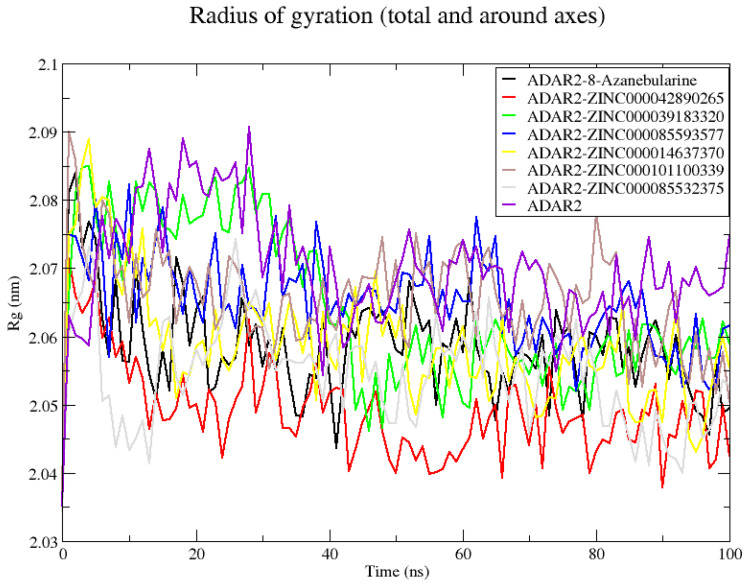
Radius of gyration (Rg) plot of the unbound ADAR2 protein and ADAR2–ligand complexes throughout the 100 ns MD simulations. On average, all the ADAR2–ligand complexes demonstrated lower Rg than the unbound ADAR2, implying higher compactness and folding.

**Figure 5 ijms-24-12612-f005:**
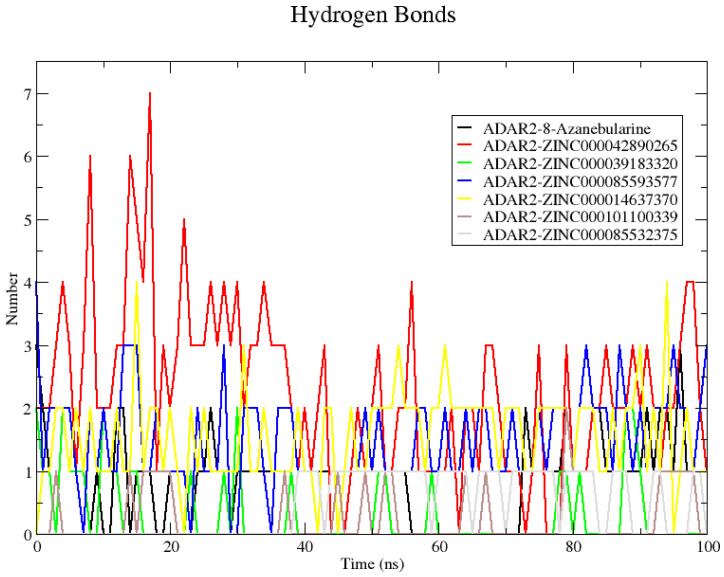
Plot showing the number of hydrogen bonds formed between ADAR2 and the shortlisted compounds throughout the 100 ns MD simulation. Compound ZINC000042890265 formed the greatest number of hydrogen bonds with ADAR2 throughout the simulation period.

**Figure 6 ijms-24-12612-f006:**
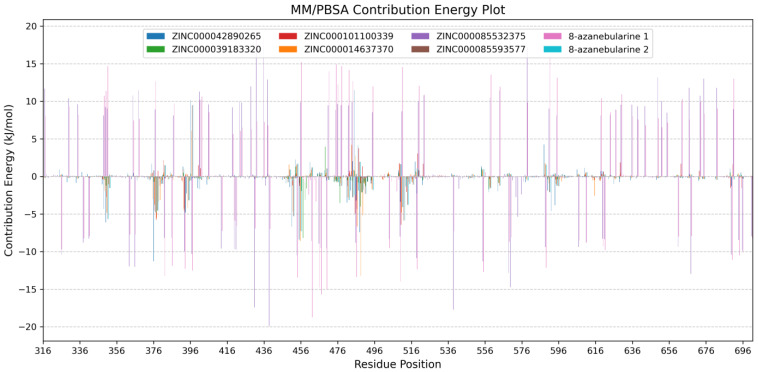
Per-residue energy decomposition plots of the ADAR2–ligand complexes. ADAR2 complexed with ZINC000042890265, ZINC000039183320, ZINC000101100339, ZINC000014637370, ZINC000085532375, ZINC000085593577, 8-azanebularine, and 8-azanebularine (re-run) are colored blue, green, red, orange, purple, brown, pink, and cyan, respectively. Grid lines are shown along the *y*-axis to help readability.

**Figure 7 ijms-24-12612-f007:**
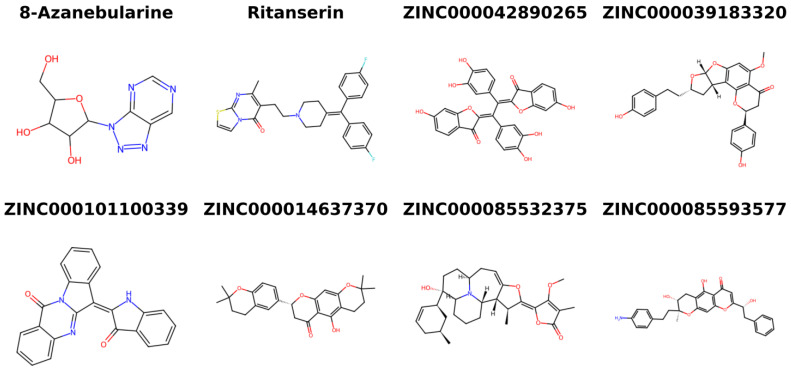
Two-dimensional structures of the known inhibitors used in this study and the shortlisted compounds.

**Table 1 ijms-24-12612-t001:** Binding energies of some TCM compounds and 8-Azanebularine after docking against the RNA binding loop of ADAR2. The interacting residues, as well as the hydrogen bond lengths, are also provided. The common names and/or IUPAC names of the compounds have been provided.

Compound	Binding Energy (kcal/mol)	Interacting Residues	
		Hydrogen Bonds (Å)	Hydrophobic Contacts
8-Azanebularine	−6.8	Thr375 (2.94, 3.14), Ile484 (2.82, 3.2), and Gly487 (3.21)	Lys376, Cys377, Ile378, His394, Lys483, and Glu485
ZINC000095913861 ((2Z)-2,11,28-trimethyl-19-methylidene-13,30- dioxaheptacyclo[21.10.1.0^6,18^.0^7,15^.0^10,14^.0^24,32^.0^27,31^]tetratriaconta- 1(33),2,6(18),7(15),10(14),11,16,23(34),24(32),27(31),28- undecaene-8,9,25,26-tetrone)	−12.0	Asn379 (2.92) and Gly489 (2.99)	Thr375, Lys376, Cys377, Ile378, His394, Arg455, Ile456, Lys483, Ile484, Glu485, Thr490, and Leu511
ZINC000085996580 (Lespedezol B2 or 8-[[2-(2,4-dihydroxyphenyl)-6-hydroxy-1-benzofuran-3-yl]methyl]-6H-[1]benzofuro[3,2-c]chromene-3,9-diol)	−11.0	Ile378 (3.01), Arg455 (2.99), Lys483 (3.17), Ile484 (2.55, 2.88), Gly487 (3.05), Leu511 (2.85), Leu512 (3.11), Thr513 (2.97), and Arg590 (2.8)	Val351, Thr375, Lys376, Cys377, His394, Thr448, Cys451, Glu485, and Leu512
ZINC000070454467 ((1S,2S,4R,6S,11R,12S,15S,18S,19S,20R,21S,23R,26S)-15-hydroxy-11,18,21-trimethyl-5,17,24,28,29-pentaoxanonacyclo[17.9.1.11,20.02,12.04,6.06,11.015,19.018,23.021,26]triacont-8-ene-10,16,25,30-tetrone)	−10.9	His394 (3.15) and Gly487 (2.91)	Thr375, Lys376, Cys377, Lys483, Ile484, Ser486, Gln488, Gly489, Arg590, and Ala595
ZINC000042890265 (Disulfuretin)	−10.6	Cys377 (3.08), Cys451 (3.15), Gly452 (3.19), Ser449 (2.7), and Arg590 (3.05)	Lys350, Val351, Thr375, Lys376, His394, Glu396, Thr448, Pro450, Arg455, Ile456, Lys483, Ile484, Gly487, Gly489, and Thr490
ZINC000039183320 (Neocalyxin A)	−10.5	Cys377 (3.01), Ile378 (3.19), Asn379 (3.11), Glu396 (2.77), and Ser449 (3.14)	Val351, Thr375, Lys376, His394, Thr448, Pro450, Cys451, Arg455, Lys483, Ile484, Glu485, Gln488, Gly489, Arg590, and Ala595
ZINC000085593577 ((2S,3R)-2-[2-(4-aminophenyl)ethyl]-3,5-dihydroxy-8-[(1R)-1-hydroxy-2-phenylethyl]-2-methyl-3,4-dihydropyrano[3,2-g]chromen-6-one)	−10.5	Lys483 (2.97), Ile484 (2.97), Gly487 (2.78), and Leu511 (2.89)	Val351, Thr375, Lys376, Cys377, Ile378, His394, Glu396, Thr448, Ser449, Cys451, Arg455, Glu485, Ser486, Gly489, Thr490, and Thr513
ZINC000070454124 ((3S,10S,11S,12S)-10,11-dihydroxy-7,18-bis(2-phenylethyl)-2,8,13,17-tetraoxapentacyclo[12.8.0.03,12.04,9.016,21]docosa-1(14),4(9),6,15,18,21-hexaene-5,20-dione)	−10.2	Lys376 (3.08), Cys377 (3.13), Asn379 (3.07), Ile484 (2.94), Ser486 (2.85), and Gly487 (3.27)	Val351, Thr375, His394, Glu396, Thr448, Ser449, Cys451, Arg455, Glu485, Ala595, Asn597, and Thr615
ZINC000103585067 ((1R,2S,5S,8S,9R,17R,18S,21S,24R,26S,27S)-5-hydroxy-2,9,26-trimethyl-3,19,23,28-tetraoxaoctacyclo[16.9.1.118,27.01,5.02,24.08,17.09,14.021,26]nonacosa-11,14-diene-4,10,22,29-tetrone)	−10.2	Thr375 (2.9), Lys376 (3.29), Cys377 (3.0), Asn379 (3.33), His394 (3.13), Ile484 (3.05), and Gly487 (2.99)	Lys483, Ser486, Gln488, and Thr615
ZINC000014637370 ((8R)-8-(2,2-dimethyl-3,4-dihydrochromen-6-yl)-5-hydroxy-2,2-dimethyl-3,4,7,8-tetrahydropyrano[3,2-g]chromen-6-one)	−10.2	Cys377 (3.34), Asn379 (2.97), Ser486 (3.31), and Gly489 (3.19)	Thr375, Lys376, His394, Cys451, Lys483, Ile484, Glu485, Gly487, and Thr490
ZINC000013384051 (Cassigarol E)	−10.1	Asn379 (3.03), Glu396 (3.13), Ser449 (2.79), and Cys451 (2.97)	Val351, Thr375, Cys377, Ile378, His394, Pro450, Arg455, Lys483, Ile484, Glu485, Gln488, Gly489, and Leu511
ZINC000059586224 ((5S)-9-methoxy-14-methyl-5,19-diphenyl-4,12,18-trioxapentacyclo[11.7.1.02,11.03,8.017,21]henicosa-1(21),2,6,8,10,13,16,19-octaen-15-one)	−10.1	Gly489 (3.07)	Thr375, Lys376, Cys377, Ile378, Asn379, His394, Arg455, Lys483, Ile484, Glu485, Gly487, Gln488, Thr490, Leu511, Asn597, and Thr615
ZINC000070454074 ((1S,2R,7R,10R,13R,14S,16R,19R,20R)-19-[(2S)-2-hydroxy-5-oxo-2H-furan-3-yl]-9,9,13,20-tetramethyl-4,15,18-trioxahexacyclo[11.9.0.02,7.02,10.014,16.014,20]docosane-5,12,17-trione)	−10.1	His394 (2.97), Arg455 (2.97, 2.96, 3.2) and Arg590 (2.81)	Val351, Thr375, Lys376, Cys377, Cys451, Ile484, Gly487, and Gln488
ZINC000085530502 ((1S,2R,4S,7S,8S,11R,12R,17S,19R,20S,24S)-19-cyclohexyl-7-(furan-3-yl)-24-hydroxy-8,19-dimethyl-3,6,14,18-tetraoxaheptacyclo[18.3.2.01,11.02,4.02,8.012,17.012,20]pentacos-21-ene-5,15,25-trione)	−10.1	Cys377 (3.22) and Arg455 (3.31)	Thr375, Lys376, His394, Cys451, Lys483, Ile484, Ser486, Gly487, Gln488, Gly489, and Thr490
ZINC000085532258 ((5E)-5-[(1S,2R,3S,11S,13S)-13-benzyl-11-[(S)-hydroxy-[(1S,5R)-5-methylcyclohex-2-en-1-yl]methyl]-3-methyl-5-oxa-10-azatricyclo[8.4.0.02,6]tetradec-6-en-4-ylidene]-3-(hydroxymethyl)-4-methoxyfuran-2-one)	−10.1	Asn379 (2.92) and Arg455 (3.32)	Val351, Thr375, Cys377, His394, Glu396, Thr448, Ser449, Cys451, Pro459, Lys483, Ile484, Glu485, Ser486, Gly487, Gln488, and Gly489
ZINC000085532442 (5-[(1S,2R,3S,4E,11S,13S)-13-benzyl-11-[(1S)-2- cyclopentyl-1-hydroxyethyl]-3-methyl-5-oxa-10- azatricyclo[8.4.0.0^2,6^]tetradec-6-en-4-ylidene]-3- (hydroxymethyl)-4-methoxy-2,5-dihydrofuran-2-one)	−10.1	Cys377 (3.04), Ile378 (2.91), and Gly489 (3.05)	Val351, Thr375, Lys376, Asn379, His394, Glu396, Ser449, Cys451, Arg455, Ile456 Lys483, Ile484, Glu485, Ser486, Gln488, and Thr490
ZINC000095911347 ((1R,2S,4R,6S,11R,12S,15R,18S,19R,20S,21S,23R,26S)-15-hydroxy-11,18,21-trimethyl-5,17,24,28,29-pentaoxanonacyclo[17.9.1.11,20.02,12.04,6.06,11.015,19.018,23.021,26]triacont-8-ene-10,16,25,30-tetrone)	−10.1	Lys376 (3.19), Cys377 (3.28), and Arg455 (2.99)	Thr375, Hs394, Ile484, Ser486, Gly487, Gln488, Gly489, and Ala595
ZINC000095914813 (5-[(Z)-2-[(2S,3S)-3-(3,5-dihydroxyphenyl)-2-(4-hydroxyphenyl)-2,3-dihydro-1-benzofuran-5-yl]ethenyl]benzene-1,3-diol)	−10.1	Gly374 (3.25), Lys376 (2.9), Cys377 (3.07), Glu396 (2.96), Ser449 (2.54), Cys451 (3.16), and Arg455 (2.8)	Lys350, Val351, Thr375, His394, Thr448, Pro450, Gly487, Gln488, Gly489, Arg590, and Ala595
ZINC000085530478 ((1S,2R,4S,7S,8S,10S,13S,17R,18S,21S,25S,27R)-7-(furan-3-yl)-25-hydroxy-8,20,20-trimethyl-3,6,15,19-tetraoxaoctacyclo[19.3.2.11,10.02,4.02,8.013,18.017,21.017,27]heptacos-22-ene-5,14,26-trione)	−10.0	Arg455 (2.81) and Gly487 (3.06)	Thr375, Lys376, Cys377, His394, Cys451, Lys483, Ile484, Ser486, and Gln488
ZINC000085530490 ((1R,2R,3’R,7S,9S,10S,12S,13S,14R,16S,19S,20S)-19-(furan-3-yl)-12-hydroxy-13,20-dimethyl-3’-propan-2-ylspiro[4,8,15,18-tetraoxahexacyclo[11.9.0.02,7.02,10.014,16.014,20]docosane-9,1’-cyclohexane]-5,11,17-trione)	−10.0	Thr375 (3.12) and Lys376 (2.89)	Cys377, His394, Cys451, Arg455, Lys483, Ile484, Ser486, Gly487, Gln488, and Gly489
ZINC000085543539 (3-[[(1R,3R)-3-[(1S,5S)-1,5-dimethylcyclohex-2-en-1-yl]cyclohexyl]methyl]-5-[(1R,4S)-4-(ethylamino)-1,2,3,4-tetrahydronaphthalen-1-yl]phenol)	−10.0	Ile484 (2.86, 3.2) and Gly487 (2.86)	Val351, Gly374, Thr375, Lys376, Cys377, His394, Ala395, Glu396, Ser449, Cys451, Arg455, Lys483, Gly489, Arg590, and Thr615
ZINC000085592995 ((1R,2R)-2-[(3S,4S)-4-hydroxy-8-[(3-hydroxyphenyl)methyl]-6-methoxy-3,4-dihydro-2H-chromen-3-yl]-1,2,3,8,9,10-hexahydropyrano[3,2-f]chromen-1-ol)	−10.0	Ser449 (2.55), Ser486 (3.19), and Gly489 (2.88)	Val351, Thr375, Cys377, His394, Glu396, Thr448, Pro450, Cys451, Arg455, Lys483, Ile484, Glu485, Gly487, Gln488, and Arg590
ZINC000085633079 (9-[[(2S,4S)-5,5-dimethyl-4’-(3-methylbut-2-enoxy)spiro[1,3-dioxolane-2,7’-furo[3,2-g]chromene]-4-yl]methoxy]furo[3,2-g]chromen-7-one)	−10.0	Lys376 (2.98), Cys377 (3.31), and Thr490 (3.07)	Ile378, His394, Arg455, Ile456, Lys483, Ile484, Glu485, Gln488, Gly489, Leu511, and Thr513
ZINC000101100339 (Qingdainone)	−9.7	Asn379 (3.01)	Lys376, Cys377, Ile378, Lys483, Ile484, Glu485, Gly487, Gly489, Thr490, Leu511, and Thr513
ZINC000085532375 ((5E)-5-[(1S,2R,3S,9S,12S,13S)-12-hydroxy-3-methyl-12-[(1S,5S)-5-methylcyclohex-2-en-1-yl]-5-oxa-17-azatetracyclo[7.7.1.02,6.013,17]heptadec-6-en-4-ylidene]-4-methoxy-3-methylfuran-2-one)	−9.6	Arg455 (3.11) and Gly487 (3.05)	Thr375, Lys376, Cys377, Asn379, His394, Ile456, Lys483, Glu485, Ser486, Gly489, and Thr490

**Table 2 ijms-24-12612-t002:** ADME prediction for some shortlisted TCM compounds.

Compound	MW (g/mol)	Consensus logP o/w	TPSA (Å^2^)	BBB Permeant	GI Absorption	ESOL Solubility Class	No. of Lipinski’s Rule Violations	No. of Veber’s Rule Violations
ZINC000095913861	556.6	5.97	94.56	No	Low	Poorly soluble	1	0
ZINC000085996580	508.48	4.47	136.66	No	Low	Poorly soluble	1	0
ZINC000070454467	526.53	4.47	137.96	No	Low	Poorly soluble	1	0
ZINC000042890265	538.46	5.27	173.98	No	High	Poorly soluble	1	0
ZINC000039183320	474.6	3.93	94.45	No	High	Moderately soluble	1	0
ZINC000085593577	487.54	3.55	126.15	No	High	Moderately soluble	0	0
ZINC000070454124	564.58	3.38	119.34	No	High	Moderately soluble	1	0
ZINC000103585067	510.53	1.27	125.43	No	High	Soluble	1	0
ZINC000014637370	408.49	4.58	64.99	Yes	High	Moderately soluble	0	0
ZINC000013384051	486.47	3.44	139.84	No	Low	Poorly soluble	1	0
ZINC000059586224	486.51	5.7	61.81	No	Low	Poorly soluble	0	0
ZINC000070454074	500.54	2.17	128.73	No	High	Soluble	1	0
ZINC000085530502	578.65	3.31	124.8	No	High	Moderately soluble	1	0
ZINC000085532258	561.71	4.27	88.46	No	High	Poorly soluble	1	0
ZINC000085532442	548.7	4.36	88.46	No	High	Poorly soluble	1	0
ZINC000095911347	526.53	1.37	137.96	No	High	Poorly soluble	1	0
ZINC000095914813	454.47	4.01	110.38	No	High	Poorly soluble	0	0
ZINC000085530478	536.57	2.32	124.8	No	High	Moderately soluble	1	0
ZINC000085530490	568.65	3.35	124.8	No	High	Moderately soluble	1	0
ZINC000085543539	471.72	7.08	32.26	No	Low	Poorly soluble	1	0
ZINC000085592995	490.54	3.47	97.61	No	High	Moderately soluble	0	0
ZINC000085633079	556.56	5.46	102.64	No	Low	Poorly soluble	1	0
ZINC000101100339	363.37	3.25	63.47	Yes	High	Moderately soluble	0	0
ZINC000085532375	481.62	3.91	68.23	Yes	High	Moderately soluble	0	0

**Table 3 ijms-24-12612-t003:** Energy terms contributing to ADAR2 binding to the top compounds and 8-Azanebularine from MM/PBSA calculations. All energy values are presented in kJ/mol as energy ± standard deviation.

Compound	vdW	Electrostatic	Polar Solvation	SASA	Binding
ZINC000042890265	−185.615 ± 1.989	−81.409 ± 2.341	174.426 ± 3.098	−24.866 ± 0.150	−117.236 ± 4.040
ZINC000039183320	−110.150 ± 2.956	−23.952 ± 1.891	82.789 ± 2.665	−15.755 ± 0.366	−67.023 ± 3.022
ZINC000101100339	−156.644 ± 1.535	12.671 ± 1.635	95.394 ± 2.438	−16.419 ± 0.171	−64.913 ± 2.029
ZINC000014637370	−218.832 ± 1.310	−42.038 ± 1.376	108.673 ± 1.974	−22.854 ± 0.099	−174.911 ± 2.104
ZINC000085532375	−69.224 ± 5.194	289.884 ± 7.174	18.341 ± 2.572	−9.868 ± 0.750	228.669 ± 3.288
ZINC000085593577	−224.512 ± 2.058	−47.130 ± 1.820	158.815 ± 2.933	−24.585 ± 0.180	−137.369 ± 2.365
8-Azanebularine	−49.600 ± 2.917	227.826 ± 11.031	75.515 ± 7.041	−7.374 ± 0.412	246.374 ± 5.841
8-Azanebularine rerun	−109.745 ± 1.390	310.967 ± 6.418	123.630 ± 5.771	−13.942 ± 0.096	310.779 ± 2.145

**Table 4 ijms-24-12612-t004:** Binding energies of top compounds and ritanserin after docking against 5-HT_2C_R. The interacting residues, as well as the hydrogen bond lengths, are also provided.

Compound	Binding Energy	Interacting Residues	
		Hydrogen Bonds (Å)	Hydrophobic Bonds
Ritanserin	−12.7	Asp134 (2.82) and Tyr358 (2.93)	Ser110, Tyr118, Val135, Ser138, Thr139, Ile142, Val208, Ser219, Ala222, Phe223, Trp324, Phe327, Phe328, Asn351, and Val354
ZINC000095913861	−12.9	-	Tyr118, Asp134, Ser138, Val208, Leu209, Val215, Ser219, Ala222, Phe223, Phe327, Phe328, Asn331, Asn351, Val354, and Tyr358
ZINC000101100339	−12.5	-	Asp134, Ser138, Leu209, Phe214, Val215, Gly218, Ser219, Ala222, Phe223, Trp324, Phe327, Phe328, and Val354
ZINC000085996580	−12.1	Ser110 (2.82), Leu209 (2.81), Ala222 (2.85), Asn331 (3.00), and Asn351 (2.78)	Ile114, Tyr118, Ile131, Asp134, Val135, Thr139, Ile142, Val208, Phe223, Trp324, Phe327, Phe328, Leu350, and Val354,
ZINC000085593577	−11.4	Asp134 (3.00, 3.06, and 3.07) and Asn331 (3.21)	Ser110, Tyr118, Trp130, Val135, Ser138, Leu209, Phe214, Val215, Ser219, Trp324, Phe327, Phe328, Val354, and Tyr358
ZINC000085532375	−11.2	Asn331 (3.06)	Asp134, Val135, Ser138, Val208, Leu209, Val215, Gly218, Ser219, Ala222, Phe223, Phe327, Phe328, Leu350, Asn351, and Val354
ZINC000014637370	−10.9	-	Asp134, Val135, Val208, Leu209, Val215, Trp324, Phe327, Phe328, Asn331, Leu350, Asn351, and Val354
ZINC000039183320	−10.3	Thr139 (2.83), Leu209 (2.92), and Asn331 (3.30)	Asp134, Val135, Ser138, Val208, Val215, Phe327, Phe328, Glu347, Leu350, Asn351, and Val354
ZINC000070454467	−8.4	Leu209 (3.01)	Asp134, Val135, Val208, Trp324, Phe327, Phe328, Asn331, Leu350, and Val354
ZINC000042890265	−7.8	Val215 (2.75) and Ser219 (2.52)	Trp130, Asp134, Val135, Ser138, Leu209, Phe214, Phe223, Trp324, Phe327, Phe328, Asn331, Leu350, and Val354

## Data Availability

Not applicable.

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
