# Peer review of "In Silico Discovery of Potential Inhibitors Targeting the RNA Binding Loop of ADAR2 and 5-HT2CR from Traditional Chinese Natural Compounds"

_ijms, 2023, doi:10.3390/ijms241612612_

Round 1
Reviewer 1 Report
In the Manuscript entitled “In Silico Discovery of Potential Inhibitors Targeting the RNA 2 Binding Loop of ADAR2 and 5-HT2CR from Traditional Chinese Natural Compounds,” the authors provide studies of molecular docking and dynamics to discover potential drugs from traditional Chinese medicine against ADAR2 and 5HT2CR. It is an exciting work, but my biggest criticism is that the authors only propose the potential of the compounds by modeling, but no experimental validation was performed. For acceptance, the authors must observe some aspects:
1 - In Table 1, only the ZINK ID of the compounds is cited, but to facilitate understanding, the name of the substance would be interesting to include. Authors are encouraged to add the substance name next to the ZINC ID.
2 – While reading the text, I had a lot of difficulty understanding the information because of the ZINC ID. The reading became rum and tedious. Thus, authors are encouraged to replace the ZINC ID with the substance's name to facilitate the readers' understanding.
3 - When looking at the MM-PBSA results, I noticed that the authors did not specify how many snapshots were used for the calculation. Whether it was 100 ns or whether it was a specific time. Authors are encouraged to specify how many snapshots were used in the calculation.
4 - When looking at the MM-PBSA results, it is noted that the binding energy for the standard ligand was (8-Azanebularine) 246,374 ± 5,841, that is, positive and irrelevant energy. In the discussion, the authors highlight, "The known ADAR2 inhibitor, 8-azanebularine, had a binding free energy of 246,374 kJ/mol (Table 3). This is not surprising as 8-azanebularine was experimentally reported to inhibit ADAR2 with a weak IC50 of 15 ± 3 mM," but positive affinity energy values mean that the ligand has no affinity for the binding site. The very high value of electrostatic energy may mean that the ligand was out of the binding site during the calculation. Authors are encouraged to redo this calculation
5 - The authors show the simulation for several complexes in the molecular dynamics simulation, but in the Per-Residue Energy Contributions, only the figure with the ADAR2-ZINC000014637370 complex is shown. The others are in the supplementary material, but adding them in a single figure would be interesting. Authors are encouraged to add others to the figure. Authors can read and cite this reference
https://www.eurekaselect.com/article/129868
6 - The authors highlight in docking that residues Thr375, Lys376, Cys377, His394, Cys451, Arg455, Lys483, 308 Ile484, Glu485, Gly487, Gln488, and Gly489 were the most common in ligand interactions. But in MMPBSA, they highlight Lys350, Cys377, Cys451, Arg455, Ser486, Gln488, and Arg510, deleting some mentioned in the docking and adding others. After all, which ones are more relevant? Authors are encouraged to discuss this, compare docking and MMPBSA interactions, and highlight which would be more important. This would generate powerful information for researchers in drug design.
7 - While reading, I noticed that the structures were not shown. Authors are encouraged to add them.
8 - Another problem is that the 2D pictures of interactions were not shown, only for the complex with 8-azanebularine and ZINC000014637370. Authors are encouraged to add them to supplemental material.
9 - In Figure 2, the 3D figure of interactions would be interesting, showing the binding mode of the most promising compound compared to the standard. Authors are encouraged to add it and discuss the interaction mode based on the compounds' chemical structure.
10 - In topic 3.5, the methodology should be better detailed. As it is, it cannot be reproduced. Also, was a web server or software used? This must be specified. In the case of a web server, authors must add the website.
12 - It would be interesting for the authors to discuss the results of each of the terms of the MM/PBSA calculations. Authors are encouraged to read and cite the reference below as a basis:
https://doi.org/10.1016/j.bmc.2021.116213
Reviewer 2 Report
The manuscript "In Silico Discovery of Potential Inhibitors Targeting the RNA Binding Loop of ADAR2 and 5-HT2CR from Traditional Chinese Natural Compounds" addresses an important issue: finding effective ADAR2 inhibitors as a potential therapeutic approach for various pathological conditions linked to aberrant RNA editing mediated by ADAR2. The authors employed a combination of computational methods, molecular dynamics simulations, binding affinity analysis, and structural similarity searches to identify and evaluate potential natural compounds from the TCM library as ADAR2 inhibitors for therapeutic applications in various diseases.
The objectives were clearly stated and explained in the manuscript, however the experimental strategy raises some major concerns and so the experimental information from which the conclusions were drawn. The manuscript is overall well written and has good organization with minor English language and style spell check required. The authors have done a great job on analyzing the experimental data and on discussing the results and their limitations, considering always different alternative explanations/considerations for interpreting the results.
The paper is interesting but there is a need for more experimental detail in order to critically review the data. Specifically, they should provide information for the following questions and comments:
Major points:
1. The authors should include more recent update on this topic and compare how this study further advances the current knowledge in the “Introduction section”.
2. Unify the style of the references in the References Section and add DOI in the cases it is possible. And use the same reference and citation (follow MDPI’s guidelines) style in the main text.
3. The Methods section in the study should be more accurately described for each technique used in the Materials & Methods Section.
4. The Conclusion Section should be more thoroughly described.
5. Caption in most of the Figures is scarce, a more detailed description is needed.
Minor points:
1. The resolution and quality of some Figures is low, the authors should provide higher quality Figures specially for Figure 3.
Round 2
Reviewer 2 Report
The authors have satisfactorily addressed the comments and recommendations provided in my initial review.